# Discretized Density-Guided Source-Free Domain Adaptation for Regression

**Gezheng Xu** [1]  **Qi Chen** [2][3]  **Qiuhao Zeng** [4][5]  **Charles Ling** [1]  **Boyu Wang** [1][5]

## Abstract

Source-Free Domain Adaptation (SFDA) enables model adaptation under distribution shifts without access to source data, providing a practical solution for privacy-sensitive applications and having shown substantial progress in classification. In contrast, regression involves ordered and continuous target variables, posing unique challenges for representation adaptation and pseudo-label refinement in the SFDA setting. To address this gap, we propose a novel algorithm for continuous label prediction in SFDA that leverages instance-dependent, discretized density–informed supervisory signals to refine pseudo-labels within an uncertainty-aware paradigm. By incorporating auxiliary discretized distribution learning, our method also promotes more compact and structured feature representations, mitigating the inherent difficulties of adapting regression models under distribution shift. We theoretically demonstrate that the resulting density structure is robust to potential perturbations, supporting reliable SFDA for regression. Extensive experiments across multiple benchmarks validate the effectiveness of the proposed approach.

## 1. Introduction

With growing needs for model adaptivity and the concerns about data privacy, Source-Free Domain Adaptation (SFDA) has emerged as a practical solution, allowing models to adapt to a target domain without requiring access to source data (Liang et al., 2020). A key focus of recent research in SFDA for classification tasks lies in constructing informative supervision signals by exploiting the geometric structure of the feature space and the intrinsic uncertainty in the output space. Specifically, the clustered structure of the feature space, induced by the discrete and non-ordinal nature of class labels, offers topological cues for refining pseudo-labels, as local neighborhoods often contain diverse supervision signals that help correct ambiguous predictions, particularly those near decision boundaries (Yang et al., 2022b; 2021; Xu et al., 2025b) (Figure 1a). Meanwhile, the softmax distribution in the output space naturally reveals sample-level uncertainty (Figure 1b), and facilitates confidence-aware self-training strategies, e.g., guiding feature alignment and further pseudo-label correction in a curriculum-based manner (Zhang et al., 2022b; Mitsuzumi et al., 2024).

By comparison, regression tasks that predict continuous labels present fundamentally different challenges. Notably, the local manifold learned by regression typically maps to smoothly varying, continuous, and inherently ordered output values. As a result, nearby points in the feature space tend to share similar predictions, offering limited supervision signals for pseudo-label refinement (Figure 1c). In addition, deep regression models are typically deterministic: the regression head outputs only a single scalar prediction for each input, providing no estimate of prediction variance or confidence. This absence of explicit uncertainty makes it difficult to assess prediction reliability or perform target value correction (Figure 1b). This issue is further exacerbated under distribution shift, during which the model's predictions become less reliable, compounding the difficulty of adaptation. Moreover, recent studies (Zhang et al., 2024; 2025) show that typical losses (e.g. MSE, $L_1$) for deep regression models tend to induce a lower marginal entropy of feature representations Z while minimizing the conditional entropy $\mathcal{H}(Z|Y)$ with respect to the target variable Y. Such entropy-reducing properties limit feature generalizability and hinder robust representation learning under inaccurate labels, making it even more challenging to adapt to the target domain in the absence of source data and target labels. Consequently, Source-Free Domain Adaptive Regression (SFDAR) remains largely underexplored and demands new approaches tailored to its unique challenges.

To this end, we propose Mutual Enhancement of Regression-Classification Integration (MERCI), a novel framework that introduces discretized density-informed supervisory signals to support SFDAR. MERCI adopts a dual-head architec-

[1]Department of Computer Science, Western University, London, Canada [2]ELLIS Institute Finland, Helsinki, Finland [3]Aalto University, Helsinki, Finland [4]University of Toronto, Toronto, Canada [5]Vector Institute, Toronto, Canada. Correspondence to: Boyu Wang <bwang@csd.uwo.ca>.

*Proceedings of the 43rd International Conference on Machine Learning*, Seoul, South Korea. PMLR 306, 2026. Copyright 2026 by the author(s).

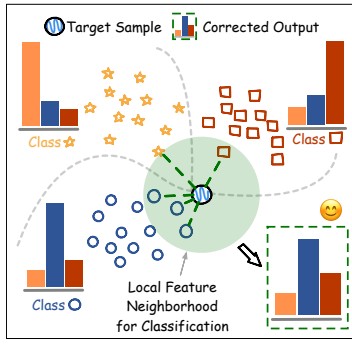

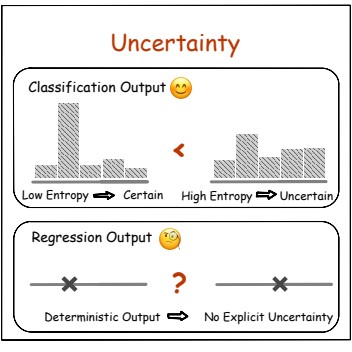

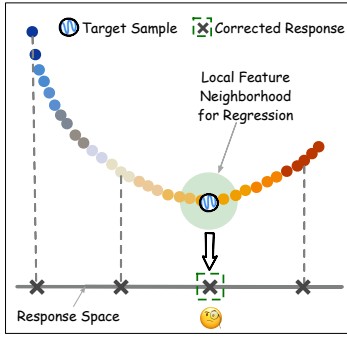

*(a)* Classification: Clustered Feature Space     *(b)* Uncertainty in Outputs     *(c)* Regression: Smooth Feature Space

*Figure 1.* This figure highlights key differences between classification and regression tasks under SFDA. (a) shows how the clustered feature space in classification supports neighborhood-based label correction, aided by distributional output information (distinct color–shape markers indicate different classes). In contrast, (c) illustrates that the smooth feature space in regression restricts the effectiveness of such refinement strategies (the smoothly changing colors represent continuous target values). (b) further demonstrates that classification outputs naturally encode uncertainty (e.g., entropy of class probabilities), while regression outputs are typically deterministic and lack explicit distributional information (e.g., entropy or confidence estimates).

ture: alongside the primary regression head, it introduces an auxiliary classification-based *histogram head* to discretize the continuous label space and parameterize a discretized conditional density of the target label. The proposed architecture can guide and regularize regression predictions through cross-head interactions, as illustrated in Figure 2. More specifically, to enable effective histogram head training, an initial uncertainty-aware label set is constructed via multiple stochastic forward passes (e.g., dropout) through the regression model. In parallel, an instance-dependent unimodal prior is imposed to reflect the inherent continuity and smoothness of each regression output. The resulting histograms enable the construction of informative, continuous supervision signals via their expected values, which in turn enhance the training of the target regression model. The benefits of incorporating discretized density signals for SFDAR are twofold. First, they introduce uncertainty-awareness into regression and provide rich distributional cues for pseudo-label refinement. Second, the distributional losses imposed by the histogram head are more effective at tightening the learned feature representations (Zhang et al., 2025), as shown in Figures 4 and E.6, thereby resolving the aforementioned fundamental challenges in SFDAR and improving adaptation performance. In addition, we theoretically show that the resulting histograms exhibit certain robustness to potential perturbations, which further supports the reliability and the effectiveness of MERCI for SFDAR.

Our main contributions can be summarized as follows:

1) We propose MERCI, a novel framework for SFDAR that integrates discretized-density estimation into regression. MERCI refines the supervision signal to guide the adaptation process by capturing the label uncertainty and further strengthens generalization by encouraging feature compression.

2) We design a structured histogram learning module that supports regression adaptation without labeled target data, and provide theoretical analysis to support its robustness and effectiveness.

3) Extensive experiments and ablation studies across diverse regression tasks demonstrate the effectiveness and reliability of MERCI. Our code is publicly available at `https://github.com/xugezheng/MERCI_SFDAR`.

## 2. Related Work

**Unsupervised Domain Adaptive Regression.** Traditional domain adaptation methods for regression typically assume access to labeled source data and unlabeled target data, aiming to improve model performance in the target domain. Mainstream approaches reduce domain discrepancy through adversarial training or explicit feature alignment (Ganin et al., 2016). For example, RSD (Chen et al., 2021) highlights the sensitivity of regression models to feature scales and minimizes subspace distances accordingly. To improve stability, DARE-GRAM (Nejjar et al., 2023) aligns the inverse Gram matrices of source and target features. Others leverage the training dynamics of distribution-informed neural networks with MMD in NTK-induced RKHS (Wu et al., 2022), or perform uncertainty-aware feature alignment (Nejjar et al., 2026). However, all these methods require access to raw source data during adaptation.

**Source-Free Domain Adaptive Regression.** To address privacy and security concerns, efforts such as TASFAR (He et al., 2024) and SSA (Adachi et al., 2025) attempt to eliminate the use of raw source data, but still depend on source-

trained modules or statistics, and thus do not fully adhere to the source-free constraint. More recently, strict Source-Free Domain Adaptive Regression (SFDAR) and related self-training paradigms have begun to emerge, marking early efforts toward regression adaptation without source data. CRAFT-U (Biswas et al., 2025), originally designed for semi-supervised learning, employs self-training strategies that can be adapted to the source-free setting but rely on assumptions or heuristics related to the target label distribution. In the context of strict SFDAR, BRR (Zhan et al., 2025) proposes a bias-reduction framework that implicitly leverages discretization to refine pseudo-labels and regulate feature representations. In contrast, our approach explicitly models the output distribution via histograms to capture both instance-level and global distributional shifts. Other related studies in distinct modalities, including energy-based test-time adaptation for depth completion (Chung et al., 2025; Wang et al., 2024), non-stationarity-aware test-time adaptation for time-series forecasting (Kim et al., 2025), and fully test-time adaptation for tabular prediction (Zhou et al., 2025), demonstrate the increasing interest in adapting models under distribution shift without source data. Although these methods offer useful insights, their architectures and problem formulations are tailored to specific tasks and cannot be directly applied to the SFDAR setting considered.

**Regression as Classification.** Recently, classification-style losses, such as cross-entropy, have been introduced into regression-related tasks, and have demonstrated promising performance (Cao et al., 2017; Sun et al., 2025; Imani & White, 2018). Zhang et al. (2023) and Imani et al. (2024) analyze the benefits of incorporating such distributional losses into regression tasks from the perspectives of feature representation and gradient stability, respectively, further validating the importance of learning a distribution over target labels, rather than predicting a single point estimate. However, these methods and theoretical findings are limited to fully supervised learning scenarios with accurate response values. In contrast, the challenge in SFDAR lies in effectively constructing meaningful classification proxies and supervision signals without source data. In our study, we propose leveraging histograms as a bridge between classification and regression to address this challenge.

## 3. Problem Setup

We consider a regression problem with input space $\mathcal{X} \subseteq \mathbb{R}^d$ and target space $\mathcal{Y} \subseteq \mathbb{R}$, where $d$ denotes the input dimension. In the SDFA setting, we assume that the source and target domain distributions, $P_{xy}^{S}$ and $P_{xy}^{T}$, are unknown and potentially different. These distributions are defined over the joint space $\mathcal{X} \times \mathcal{Y}$, and can be factorized into marginal and conditional components as follows: $P_{xy}^{S} = P_{x}^{S} P_{y|x}^{S}$ and $P_{xy}^{T} = P_{x}^{T} P_{y|x}^{T}$, respectively. In SFDA, we are

given a *source regression model* $h_{S} : \mathcal{X} \to \mathcal{Y}$ pretrained on a *labeled source dataset* $\mathcal{D}_{S} \triangleq \{x_i^s, y_i^s\}_{i=1}^{N_S}$ sampled from $P_{xy}^{S}$, along with an *unlabeled target dataset* $\mathcal{D}_{T} \triangleq \{x_i^T\}_{i=1}^{N_T}$ drawn from $P_{x}^{T}$. Given access to $h_{S}$ and $\mathcal{D}_{T}$, our objective is to learn a *target regression model* $h_{T} : \mathcal{X} \to \mathcal{Y}$ that performs well on the target domain by adapting $h_{S}$ on $\mathcal{D}_{T}$, without access to the original source data $\mathcal{D}_{S}$.

We denote the source and target models as $h_{S} = f_{\text{reg},S} \circ g_{S}$ and $h_{T} = f_{\text{reg},T} \circ g_{T}$, where $g_{S}$ and $g_{T}$ are feature extractors, $f_{\text{reg},S}$ and $f_{\text{reg},T}$ represent the corresponding regression heads, and $\circ$ denotes the functional composition operator. In practice, the source and target models $h_{S}$ and $h_{T}$ typically share the same network architecture, with the target model initialized from the source model. We therefore use the generic notation $h = f_{\text{reg}} \circ g$ to refer to either model unless a distinction is necessary.

**Notations.** For any $v \in \mathbb{R}^q$, we let $\|v\| \triangleq \sqrt{\sum_{j=1}^{q} v_j^2}$ denote its $\ell_2$ norm. For any positive integer $K$, we denote $[K] \triangleq \{1, \dots, K\}$, and define the $(K-1)$-dimensional probability simplex as: $\Delta^{K-1} \triangleq \{\mathbf{s} \in \mathbb{R}^K : s_j \geq 0, \sum_{k=1}^{K} s_j = 1\}$. For any $p \in \Delta^{K-1}$, we also use $p[j]$ to denote its $j$th component for $j \in [K]$. We use capital letters (e.g., $X^T$, $Y^T$) to denote random variables, and the corresponding lowercase letters (e.g., x, y, $x^T$, $y^T$) to denote their realizations. For any discrete set $\mathcal{S}$, we use $|\mathcal{S}|$ to denote its cardinality.

## 4. Methodology

An overview of our framework, MERCI, is shown in Figure 2. MERCI employs a dual-head architecture consisting of a regression head ($f_{\text{reg}}$) and a classification-based histogram head ($f_{\text{hist}}$), both built upon a shared feature extractor ($g$). Importantly, the histogram head in our framework is not merely used for conventional classification: it approximates the discretized density of the target label $Y^T$ given input $X^T = x$. The framework operates via bi-directional knowledge transfer between the two heads through a self-adaptive histogram structure, with the regression-to-classification and classification-to-regression processes detailed in Sections 4.1 and 4.2, respectively.

By leveraging the *ordered* and *continuous* nature of regression outputs, this design enables better uncertainty characterization and facilitates more robust adaptation in the absence of source data and target label. Moreover, the loss associated with the histogram head implicitly encourages the learned features to compress irrelevant label information, which in turn improves cross-domain generalization (Zhang et al., 2025).

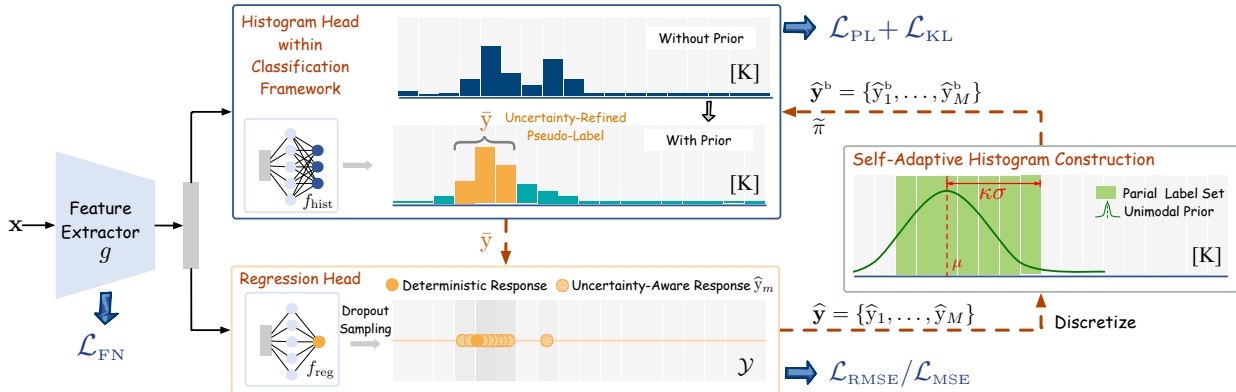

*Figure 2. Overview of MERCI. The regression head $f_{\text{reg}}$ produces multiple (M) dropout-based stochastic predictions, from which the instance-level mean $\mu$ and standard deviation $\sigma$ are estimated. These statistics define a unimodal prior with variance scaling factor $\kappa$, and the stochastic prediction set $\hat{\mathbf{y}}$ is discretized into $\hat{\mathbf{y}}^b$ to supervise the histogram head. The histogram head $f_{\text{hist}}$ refines these supervision signals via $\mathcal{L}_{\text{PL}}$ and $\mathcal{L}_{\text{KL}}$, and the resulting refined continuous pseudo-labels $\bar{\mathbf{y}}$ are fed back to guide $f_{\text{reg}}$ through the regression losses $\mathcal{L}_{\text{RMSE}}$ or $\mathcal{L}_{\text{MSE}}$. The red dashed arrows in the figure illustrate the cross-head interactions within the MERCI framework.*

## 4.1. Regression-to-Classification: Histogram-Based Posterior Estimation

We begin by describing how to leverage the regression model's predictions to guide the training of the histogram head and obtain a discrete approximation of the conditional distribution $\mathbb{P}(Y^{\text{T}}|X^{\text{T}} = \mathbf{x})$. Specifically, let $f_{\text{hist}}$ denote the histogram head applied after the shared feature extractor $g$, such that $h_{\text{hist}} \triangleq f_{\text{hist}} \circ g : \mathcal{X} \to \Delta^{K-1}$ maps each input to a $K$-way discrete probability distribution, referred as a *histogram* throughout the paper. Here, the label space $\mathcal{Y}$ is uniformly partitioned into $K$ bins of equal width: $\mathcal{Y} = \cup_{k=1}^{K} \mathcal{Y}_k$, where the number of bins $K$ is automatically selected by the algorithm. Under this binning scheme, the $k$th element of the histogram head output, $h_{\text{hist}}(\mathbf{x})[k]$, approximates the probability that the target label falls within bin $\mathcal{Y}_k$, i.e., $\mathbb{P}(Y^{\text{T}} \in \mathcal{Y}_k | X^{\text{T}} = \mathbf{x})$. For each bin $\mathcal{Y}_k$, we associate a representative value $\widetilde{y}_k$, such as the left endpoint, right endpoint, or midpoint of the interval. We denote the collection of all representative values as $\widetilde{\mathcal{Y}} \triangleq \{\widetilde{y}_1, \ldots, \widetilde{y}_K\}$, with each $\widetilde{y}_k$ corresponding uniquely to a class index (i.e., bin index) $k$ in the discrete label space $[K]$. We have included the details of selecting the bin number K and assigning the bin representative values in the Appendix C.1.

**Histogram Approximation.** To capture predictive uncertainty for a given input $\mathbf{x}$ from the target domain, we perform $M$ stochastic forward passes through the source regression model by enabling dropout at inference time. This yields an *ensemble of predictions* $\hat{\mathbf{y}} \triangleq \{\hat{y}_1, \ldots, \hat{y}_M\}$, which are used to construct a partial label set to supervise the learning of the histogram head. To further refine the approximation of the discretized conditional density of the target label, we incorporate the inherent continuity and ordering of regression outputs. In many real-world settings, such as age estimation, house price prediction, or angle estimation, the label distribution conditioned on a given input tends to be unimodal,

with the target value concentrated around a plausible mean and exhibiting smooth variation (Biswas et al., 2025). To reflect this, we impose a *unimodal prior* on the conditional label distribution $\pi(y|\mathbf{x})$ defined over label space $\mathcal{Y}$. For simplicity, in our implementation, we construct a Gaussian prior for $\pi$, as detailed in Appendices C.1 and E.8.

To integrate the unimodal prior into the histogram-based approximation, we compute the probability mass of $\pi$ over the $K$ predefined bins and use these values as a discrete prior distribution on $\widetilde{\mathcal{Y}}$, denoted as $\widetilde{\pi}$. Additionally, let $\hat{\mathbf{y}}^b \triangleq \{\hat{y}_1^b, \ldots, \hat{y}_M^b\}$ denote the discrete label set obtained by mapping the original continuous predictions $\hat{\mathbf{y}}$ to their corresponding bin indices. By combining the uncertainty-aware partial label set $\hat{\mathbf{y}}^b$ with the unimodal prior belief $\widetilde{\pi}$, we approximate the histogram distribution for the unknown target label $Y^{\text{T}}$ through the following optimization problem:

$$\widetilde{p}^{\star} \in \arg\inf_{\widetilde{p} \in \Pi} \left\{ \ell_{\text{PL}}(\widetilde{p}; \hat{\mathbf{y}}^b) + \lambda_{\text{prior}} \mathsf{d}(\widetilde{p}, \widetilde{\pi}) \right\}. \quad (1)$$

Here, $\Pi \subseteq \Delta^{K-1}$ denotes a parameterized set of discrete distributions over $K$ classes, induced by the hypothesis space of the histogram model $h_{\text{hist}}$, and $\lambda_{\text{prior}}$ is a learnable coefficient. For ease of presentation, we assume that $\Pi$ is a convex and closed set; otherwise, we replace it with its closed convex hull in (1), as defined in Definition B.4 and further discussed in Remark B.5. The function $\ell_{\text{PL}}(\widetilde{p}; \hat{\mathbf{y}}^b)$ represents the loss associated with the partial label set $\hat{\mathbf{y}}^b$, and we assume that the mapping $\widetilde{p} \mapsto \ell_{\text{PL}}(\widetilde{p}; \hat{\mathbf{y}}^b)$ is convex and lower semicontinuous; in our implementation, we instantiate $\ell_{\text{PL}}$ using the partial label loss defined in (C3). The divergence function $\mathsf{d}(\cdot, \cdot)$ is defined in Definition B.1.

**Interpretation.** The optimization problem in (1) admits a twofold interpretation. First, it can be viewed as a generalization of Bayesian inference, characterized by three key arguments: (a) a loss function $\ell_{\text{PL}}$, (b) a divergence $\mathsf{d}$

measuring the deviation from a prior $\widetilde{\pi}$, and (c) a feasible solution space $\Pi$. From this perspective, the optimal solution $\widetilde{p}^\star$ to (1) can be interpreted as an extension to the generalized variational Bayesian posterior (Knoblauch et al., 2019; Husain & Knoblauch, 2022; Soen et al., 2024), balancing empirical performance with adherence to prior knowledge. The Bayesian interpretation of (1) is further discussed in Appendix B.2.

Beyond the Bayesian viewpoint, we also establish a robustness property associated with $\widetilde{p}^\star$, highlighting its resilience to certain perturbations. We formalize this connection in the following proposition.

**Proposition 4.1.** *Let $\mathtt{d}_{\widetilde{\pi}}^*(\cdot)$ denote the the Legendre-Fenchel conjugate of $\mathtt{d}(\cdot, \widetilde{\pi})$, as defined in Definition B.3. Let $\ell^\star$ represent a maximizer of the optimization problem:*

$$\ell^\star \in \underset{\ell \in \mathcal{F}_b([K])}{\arg\sup} \left\{ \inf_{\widetilde{p} \in \Pi} [\ell_{\mathrm{PL}}(\widetilde{p}; \widehat{\mathbf{y}}^{\mathrm{b}}) + \mathbb{E}_{\widetilde{p}}\{\ell(Y^{\mathrm{T}})\}] \\ - \lambda_{\mathrm{prior}} \mathtt{d}_{\widetilde{\pi}}^* \left( \frac{\ell}{\lambda_{\mathrm{prior}}} \right) \right\}, \quad (2)$$

*where $\mathcal{F}_b([K])$ denotes the set of all bounded and measurable functions mapping from the discrete label space $[K]$ to $\mathbb{R}$. Then, for the optimal solution $\widetilde{p}^\star$ to (1), the following holds:*

$$\widetilde{p}^\star \in \underset{\widetilde{p} \in \Pi}{\arg\inf} [\ell_{\mathrm{PL}}(\widetilde{p}; \widehat{\mathbf{y}}^{\mathrm{b}}) + \mathbb{E}_{\widetilde{p}}\{\ell^\star(Y^{\mathrm{T}})\}]. \quad (3)$$

The proof of Proposition 4.1 is provided in Appendix B.2. According to Proposition 4.1, the optimal solution $\widetilde{p}^\star$ to (1) minimizes the loss under perturbations introduced by an adversary. This implies that solving (1) to approximate the histogram distribution yields the optimal belief even when the original loss function $\ell_{\mathrm{PL}}$ is replaced by a perturbed version, thereby ensuring robustness to potential distribution shift and model misspecification in the SFDA setting.

## 4.2. Classification-to-Regression: Histogram Information Transfer

We next describe two strategies that make use of the approximated discretized density (histogram) to supervise the learning of the regression head.

**Pseudo-Label Calculation.** For each $\mathbf{x}_i^{\mathrm{T}}$ in the target domain data $\mathcal{D}_{\mathrm{T}}$, let $\widetilde{p}_i^{\mathrm{old}} \in \Delta^{K-1}$ denote the approximated histogram distribution produced by the histogram head $h_{\mathrm{hist}}$. To better stabilize the adaptation process and retain useful information from the source model's predictions (Qiu et al., 2021), we update $\widetilde{p}_i^{\mathrm{old}}$ using a moving average strategy: $\widetilde{p}_i^{\mathrm{old}} = \mathtt{NORMALIZE}\{\alpha \widetilde{p}_i^{\mathrm{old}} + (1 - \alpha)\widehat{p}_i\}$. To obtain a more confident pseudo-label, we compute a truncated expectation over the most probable bins in $\widetilde{p}_i^{\mathrm{old}}$, effectively filtering out uncertain predictions. Specifically, we start

from the bin with the highest predicted probability and sequentially include bins in descending order of probability until the cumulative mass reaches a predefined confidence mass threshold $\tau$. Let $\mathcal{I}_i \subseteq [K]$ denote the minimal such subset of bins, formally defined as:

$$\mathcal{I}_i \in \underset{\mathcal{S} \subseteq [K]}{\arg\min} |\mathcal{S}| \ \text{ s.t. } \sum_{j \in \mathcal{S}} \widetilde{p}_i^{\mathrm{old}}[j] \geq \tau.$$

The continuous pseudo-label is then computed as a normalized weighted average over the representative values $\widetilde{y}_j$ corresponding to the selected bins:

$$\bar{\mathrm{y}}_i \triangleq \frac{1}{\mathtt{C}} \sum_{j \in \mathcal{I}_i} \widetilde{p}_i^{\mathrm{old}}[j] \cdot \widetilde{y}_j, \ \text{ where } \mathtt{C} = \sum_{j \in \mathcal{I}_i} \widetilde{p}_i^{\mathrm{old}}[j]. \quad (4)$$

During training, these uncertainty-refined pseudo-labels are used to supervise the learning of the regression head by minimizing the following batch-wise mean squared error (MSE) over a batch $\mathcal{B}$:

$$\mathcal{L}_{\mathrm{MSE}} \triangleq \frac{1}{|\mathcal{B}|} \sum_{i \in \mathcal{B}} \{h_{\mathrm{reg}}(\mathbf{x}_i^{\mathrm{T}}) - \bar{\mathrm{y}}_i^{\mathrm{T}}\}^2. \quad (5)$$

**Robust Loss Version for Linear Regression.** In addition to the standard MSE loss, we further account for the uncertainty inherent in the pseudo-labels, which is an important consideration in the SFDA setting due to domain shifts. To this end, we adopt a distributionally robust optimization (DRO) objective that minimizes the worst-case risk over a neighborhood of the empirical distribution. Specifically, let $\mathbf{z}_i^{\mathrm{T}} \triangleq g(\mathbf{x}_i^{\mathrm{T}})$ denote the feature representation of input $\mathbf{x}_i^{\mathrm{T}}$ obtained from the shared extractor. We model the regression head $f_{\mathrm{reg}}$ as a linear function: $y = \langle \beta, z^{\mathrm{T}} \rangle$, where $\langle \cdot, \cdot \rangle$ denotes the inner product, and $\beta$ is a weight vector that incorporates both weights and bias terms. We define the pointwise MSE as: $\ell_{\mathrm{MSE}}(z, y; \beta) = (\langle \beta, z \rangle - y)^2$. Let $\widehat{p}_{N_{\mathrm{T}}}$ denote the empirical distribution over the pseudo-label-feature pairs $\{(\mathbf{z}_i^{\mathrm{T}}, \bar{y}_i)\}_{i=1}^{N_{\mathrm{T}}}$. To incorporate pseudo-label uncertainty, we minimize the following worst-case expected loss over all distributions within a divergence ball of radius $\delta$ centered at $\widehat{p}_{N_{\mathrm{T}}}$:

$$\inf_{\beta \in \mathbb{R}^r} \sup_{p: \ \mathtt{W}_2(p, \widehat{p}_{N_{\mathrm{T}}}) \leq \delta} \mathbb{E}_p\{\ell_{\mathrm{MSE}}(\mathbf{Z}^{\mathrm{T}}, \bar{\mathbf{Y}}^{\mathrm{T}}; \beta)\}, \quad (6)$$

where $r$ denotes the feature dimension, $\mathtt{W}_2(\cdot, \cdot)$ represents the optimal transport divergence with square cost function, as defined in Definition B.2, and the expectation in (6) is taken with respect to the joint distribution of $(\mathbf{Z}^{\mathrm{T}}, \bar{\mathbf{Y}}^{\mathrm{T}})$ induced by $p$.

**Lemma 4.2.** *(Blanchet et al., 2019) The optimal solution $\beta^\star$ to the worst-case risk minimization problem in (6) can be equivalently obtained by solving the following regularized objective:*

$$\beta^\star \in \underset{\beta \in \mathbb{R}^r}{\arg\inf} \left\{ \mathcal{L}_{\mathrm{RMSE}}(\beta; \{(\mathbf{z}_i^{\mathrm{T}}, \bar{y}_i)\}_{i=1}^{N_{\mathrm{T}}}) + \sqrt{\delta}\|\beta\|_2 \right\},$$

*Table 1.* **MAE** and **$R^2$** on *Biwi-Kinect* dataset. *Best non-Oracle SF results are* **bolded**, *best non-Oracle results are* underlined.

| Method | Type | Female → Male (MAE) | | | | Male → Female (MAE) | | | | Female → Male (R²) | | | | Male → Female (R²) | | | |
|---|---|---|---|---|---|---|---|---|---|---|---|---|---|---|---|---|---|
| | | Pitch | Roll | Yaw | Mean | Pitch | Roll | Yaw | Mean | Pitch | Roll | Yaw | Mean | Pitch | Roll | Yaw | Mean |
| DANN (Ganin et al., 2016) | UDA | 4.6776 | 4.6790 | 3.4183 | 4.2583 | 7.0520 | 5.0633 | 6.3409 | 6.1521 | 0.9213 | 0.6011 | 0.9764 | 0.8329 | 0.8674 | 0.6387 | 0.9292 | 0.8118 |
| RSD (Chen et al., 2021) | UDA | 4.7010 | 4.7304 | 3.3952 | 4.2755 | 7.0438 | 4.9587 | 6.0774 | 6.0266 | 0.9217 | 0.6060 | 0.9765 | 0.8347 | 0.8586 | 0.6542 | 0.9347 | 0.8158 |
| DARE-GRAM (Nejjar et al., 2023) | UDA | 4.8748 | 4.8832 | 3.4692 | 4.4091 | 6.9178 | 5.2421 | 5.5510 | 5.9036 | 0.9188 | 0.5829 | 0.9753 | 0.8257 | 0.8679 | 0.6414 | 0.9439 | 0.8178 |
| SSA (Adachi et al., 2025) | DF | 5.0149 | 4.9848 | 3.5495 | 4.5164 | 7.7513 | 5.1466 | 6.2680 | 6.3886 | 0.9092 | 0.5616 | 0.9748 | 0.8152 | 0.8659 | 0.6176 | 0.9298 | 0.8045 |
| TASFAR (He et al., 2024) | DF | 5.0718 | 5.0165 | 3.7800 | 4.6228 | 8.5715 | 5.6320 | 6.1813 | 6.7949 | 0.9081 | 0.5346 | 0.9724 | 0.8050 | 0.8034 | 0.5647 | 0.9048 | 0.7576 |
| Source | SF | 5.1743 | 5.0695 | 3.6009 | 4.6149 | 8.7046 | 5.3419 | 6.3284 | 6.7916 | 0.9092 | 0.5459 | 0.9748 | 0.8100 | 0.8049 | 0.5752 | 0.9260 | 0.7687 |
| BN-adapt (Benz et al., 2021) | SF | 5.0398 | 5.0036 | 3.5446 | 4.5293 | 8.2665 | 5.2135 | 6.0984 | 6.5261 | 0.9069 | 0.5502 | 0.9751 | 0.8107 | 0.8267 | 0.6059 | 0.9332 | 0.7886 |
| CRAFT-U (Biswas et al., 2025) | SF | 5.0468 | 5.0186 | 3.5374 | 4.5343 | 8.1190 | 5.2070 | 6.0558 | 6.4606 | 0.9063 | 0.5461 | 0.9753 | 0.8093 | 0.8310 | 0.6074 | 0.9337 | 0.7907 |
| BRR (Zhan et al., 2025) | SF | 4.9856 | 4.9647 | 3.5361 | 4.4954 | 8.1646 | 5.1634 | 6.0874 | 6.4718 | 0.9061 | 0.5603 | 0.9753 | 0.8139 | 0.8319 | 0.5993 | 0.9329 | 0.7880 |
| VM | SF | 4.8403 | 4.9430 | 3.4149 | 4.3994 | 8.4844 | 5.1861 | 6.1158 | 6.5955 | 0.9148 | 0.5597 | 0.9767 | 0.8171 | 0.8124 | 0.6048 | 0.9299 | 0.7824 |
| AugSelfTr | SF | 4.9817 | 4.9731 | 3.4001 | 4.4517 | 7.8359 | 5.0880 | 5.4812 | 6.1350 | 0.9108 | 0.5586 | 0.9769 | 0.8154 | 0.8288 | 0.6231 | 0.9410 | 0.7976 |
| MERCI | SF | 4.6454 | 4.9101 | 3.4350 | 4.3302 | 7.1005 | 5.1010 | 5.0537 | 5.7517 | 0.9212 | 0.5642 | 0.9767 | 0.8207 | 0.8441 | 0.6218 | 0.9527 | 0.8062 |
| MERCI w. FN | SF | 4.6464 | 4.9037 | 3.4126 | 4.3209 | **7.0032** | 5.0866 | 4.6776 | 5.5892 | 0.9222 | 0.5718 | 0.9771 | 0.8237 | **0.8502** | 0.6281 | 0.9594 | 0.8126 |
| MERCI-R | SF | **4.5074** | 4.8546 | **3.3719** | **4.2446** | 7.4265 | 5.1209 | 5.1885 | 5.9120 | **0.9249** | **0.5745** | 0.9775 | 0.8256 | 0.8373 | 0.6235 | 0.9502 | 0.8037 |
| MERCI-R w. FN | SF | 4.5551 | **4.8520** | 3.3933 | 4.2668 | 7.0206 | **5.0580** | **4.4793** | **5.5193** | 0.9226 | 0.5717 | 0.9775 | 0.8239 | 0.8490 | **0.6362** | **0.9630** | **0.8161** |
| Oracle | - | 0.8474 | 1.0630 | 0.8644 | 0.9249 | 1.1487 | 1.3535 | 1.0908 | 1.1977 | 0.9973 | 0.9805 | 0.9984 | 0.9921 | 0.9963 | 0.9767 | 0.9978 | 0.9903 |

*where* $\mathcal{L}_{\mathrm{RMSE}}(\beta; \{(z_i^{\mathrm{T}}, \bar{y}_i)\}_{i=1}^{N_{\mathrm{T}}}) \triangleq \sqrt{\frac{1}{N_{\mathrm{T}}} \sum_{i=1}^{N_{\mathrm{T}}} (\langle \beta, z_i \rangle - \bar{y}_i)^2}$, *referred to as root mean square error (RMSE).*

As indicated by Lemma 4.2, once the pseudo-labels $\bar{y}_i$ are obtained, a solution that is robust to potential uncertainty can be derived by solving a regularized risk minimization problem. Specifically, the distributionally robust solution corresponds to minimizing the RMSE with an $\ell_2$ regularization term (i.e., weight decay). Therefore, we optimize the regression head by minimizing the following batch-wise RMSE objective:

$$\mathcal{L}_{\mathrm{RMSE}} = \sqrt{\mathcal{L}_{\mathrm{MSE}}} = \sqrt{\frac{1}{|\mathcal{B}|} \sum_{i \in \mathcal{B}} \left\{ h_{\mathrm{reg}}(x_i^{\mathrm{T}}) - \bar{y}_i^{\mathrm{T}} \right\}^2}. \quad (7)$$

### 4.3. Overall Training Objective

For each input $x_i^{\mathrm{T}}$ in a batch $\mathcal{B}$, let $\widehat{\mathbf{y}}_i^{\mathrm{b}} \triangleq \{\widehat{y}_{i,1}^{\mathrm{b}}, \ldots, \widehat{y}_{i,M}^{\mathrm{b}}\}$ denote the discrete labels obtained by mapping the continuous predictions to the associated bin indices; $\widetilde{p}_i = h_{\mathrm{hist}}(x_i^{\mathrm{T}}) \in \Delta^{K-1}$ the current histogram output; $\widetilde{p}_i^{\mathrm{old}}$ the histogram obtained from previous training epochs; and $\widetilde{\pi}_i$ the prior constructed from $\widetilde{p}_i^{\mathrm{old}}$ and $\widehat{\mathbf{y}}_i^{\mathrm{b}}$. We define the following batch-wise loss terms:

$$\mathcal{L}_{\mathrm{KL}} = \frac{1}{|\mathcal{B}|} \sum_{i \in \mathcal{B}} \ell_{\mathrm{KL}}(\widetilde{p}_i; \widehat{\mathbf{y}}_i^{\mathrm{b}}, \widetilde{p}_i^{\mathrm{old}}), \mathcal{L}_{\mathrm{PL}} = \frac{1}{|\mathcal{B}|} \sum_{i \in \mathcal{B}} \ell_{\mathrm{PL}}(\widetilde{p}_i; \widehat{\mathbf{y}}_i^{\mathrm{b}}),$$

where $\ell_{\mathrm{KL}}(\widetilde{p}_i; \widehat{\mathbf{y}}_i^{\mathrm{b}}, \widetilde{p}_i^{\mathrm{old}}) = d_{\mathrm{KL}}(\widetilde{p}, \widetilde{\pi}) = \sum_{k=1}^{K} \widetilde{\pi}_i[k] \cdot \log\left(\frac{\widetilde{\pi}_i[k]}{\widetilde{p}_i[k]}\right)$, $\ell_{\mathrm{PL}}(\widetilde{p}_i; \widehat{\mathbf{y}}_i^{\mathrm{b}}) \triangleq -\frac{1}{|\widehat{\mathbf{y}}_i^{\mathrm{b}}|} \sum_{k \in \widehat{\mathbf{y}}_i^{\mathrm{b}}} \log \widetilde{p}_i[k]$, and $\mathcal{L}_{\mathrm{PL}}$ and $\mathcal{L}_{\mathrm{KL}}$ correspond to the partial-label and the prior-regularization terms in (1), respectively. In addition, to account for the role of feature scale in regression (Chen et al., 2021), we optionally include a feature-norm consistency term: $\mathcal{L}_{\mathrm{FN}} \triangleq \frac{1}{|\mathcal{B}|} \sum_{i \in \mathcal{B}} \left| \|g(x_i^{\mathrm{T}})\| - \|g_{\mathrm{S}}(x_i^{\mathrm{T}})\| \right|$.

Consequently, the final objective, depending on whether $\mathcal{L}_{\mathrm{MSE}}$ in Eq. (5) or $\mathcal{L}_{\mathrm{RMSE}}$ in (7) is used, is given by:

$$\mathcal{L}_{\mathrm{MERCI}} = \lambda_{\mathrm{PL}} \mathcal{L}_{\mathrm{PL}} + \lambda_{\mathrm{prior}} \mathcal{L}_{\mathrm{KL}} + \lambda_{\mathrm{MSE}} \mathcal{L}_{\mathrm{MSE}} \ (+\lambda_{\mathrm{FN}} \mathcal{L}_{\mathrm{FN}});$$
$$\mathcal{L}_{\mathrm{MERCI-R}} = \lambda_{\mathrm{PL}} \mathcal{L}_{\mathrm{PL}} + \lambda_{\mathrm{prior}} \mathcal{L}_{\mathrm{KL}} + \lambda_{\mathrm{RMSE}} \mathcal{L}_{\mathrm{RMSE}} \ (+\lambda_{\mathrm{FN}} \mathcal{L}_{\mathrm{FN}}),$$

where $\lambda_{\mathrm{PL}}$, $\lambda_{\mathrm{prior}}$, $\lambda_{\mathrm{MSE}}/\lambda_{\mathrm{RMSE}}$, and $\lambda_{\mathrm{FN}}$ are learnable weighting coefficients.

## 5. Experimental Results

**Datasets and Evaluation Metrics.** We conduct experiments on four regression tasks: **age estimation** on UTK-Face (Zhang et al., 2017), **head pose estimation** on Biwi-Kinect (Fanelli et al., 2013), **house price prediction** on the California Housing dataset (Pace & Barry, 1997), and **digit prediction** on two digit datasets, SVHN (Netzer et al., 2011) and MNIST (LeCun et al., 1998). For the UTKFace and Biwi-Kinect datasets, we use gender as the domain attribute and consider two domains, Female and Male, for adaptation. In the California Housing dataset, we treat different geographic regions (Near Bay and Far Bay) as distinct domains. For the digit datasets, we follow the setup in Adachi et al. (2025), and adaptation is performed across different datasets. To comprehensively evaluate the performance of our regression models, we employ four widely used evaluation metrics: Mean Absolute Error (MAE), Root Mean Squared Error (RMSE), Coefficient of Determination ($R^2$), and Pearson Correlation Coefficient (R). **Methods.** As a relatively underexplored research topic, Source-Free Domain Adaptation for Regression (SFDAR) lacks sufficient established baselines. To enable a comprehensive evaluation, we compare our approach with representative methods from three related settings: Unsupervised Domain Adaptation (UDA), Data-Free (DF) Domain Adaptation, and Source-Free (SF) Domain Adaptation. For the UDA setting, we adopt DANN (Ganin et al., 2016), RSD (Chen et al., 2021), and DARE-GRAM (Nejjar et al., 2023). For the data-free setting, we consider SSA (Adachi et al., 2025) and TAS-FAR (He et al., 2024), which require no raw source data but rely on source-derived statistics or pre-trained components. For the source-free setting, we compare against test-time Batch Normalization (BN-adapt) (Benz et al., 2021), CRAFT-U (Biswas et al., 2025), and BRR (Zhan et al., 2025), alongside two augmentation-based baselines adapted

*Table 2. Results on **UTKFace (Left)** and **California Housing (Right)** datasets. Best non-Oracle SF results are **bolded**, best non-Oracle results are underlined.*

| Method | Type | Female → Male | | | | Male → Female | | | | Far Bay → Near Bay | | | | Near Bay → Far Bay | | | |
|---|---|---|---|---|---|---|---|---|---|---|---|---|---|---|---|---|---|
| | | MAE ↓ | RMSE ↓ | R² ↑ | R ↑ | MAE ↓ | RMSE ↓ | R² ↑ | R ↑ | MAE ↓ | RMSE ↓ | R² ↑ | R ↑ | MAE ↓ | RMSE ↓ | R² ↑ | R ↑ |
| DANN (Ganin et al., 2016) | UDA | 6.2233 | 8.7264 | 0.8088 | 0.9005 | 7.0306 | 10.3967 | 0.7295 | 0.8623 | 0.4907 | 0.6957 | 0.5919 | 0.8129 | 0.5605 | 0.7196 | 0.4811 | 0.7365 |
| RSD (Chen et al., 2021) | UDA | 6.2172 | 8.7435 | 0.8080 | 0.9007 | 7.0332 | 10.3930 | 0.7298 | 0.8624 | – | – | – | – | – | – | – | – |
| DARE-GRAM (Nejjar et al., 2023) | UDA | 6.2622 | 8.8235 | 0.8045 | 0.8986 | 6.9333 | 10.2150 | 0.7390 | 0.8678 | 0.4776 | 0.6762 | 0.6144 | 0.8200 | 0.5678 | 0.7305 | 0.4647 | 0.7263 |
| SSA (Adachi et al., 2025) | DF | 7.0709 | 10.1284 | 0.7423 | 0.8861 | 7.6017 | 10.6361 | 0.7170 | 0.8530 | 0.4690 | 0.6550 | 0.6390 | 0.8017 | 0.7711 | 1.0032 | -0.0065 | 0.4088 |
| TASFAR (He et al., 2024) | DF | 7.2788 | 9.7731 | 0.7656 | 0.8804 | 10.1191 | 16.6058 | 0.3070 | 0.6634 | 0.5055 | 0.6714 | 0.6247 | 0.7975 | 0.7366 | 0.9179 | 0.1581 | 0.5095 |
| Source | SF | 6.5494 | 9.1600 | 0.7892 | 0.8998 | 7.9390 | 12.4863 | 0.6098 | 0.8369 | 0.5160 | 0.6840 | 0.6050 | 0.7899 | 0.7346 | 0.9039 | 0.1830 | 0.5678 |
| BN-adapt (Benz et al., 2021) | SF | 6.3018 | 9.0182 | 0.7957 | 0.8951 | 7.3387 | 10.7872 | 0.7090 | 0.8581 | 0.6644 | 0.8767 | 0.3519 | 0.7794 | 0.7068 | 0.8840 | 0.2174 | **0.6700** |
| CRAFT-U (Biswas et al., 2025) | SF | 6.2393 | 8.9087 | 0.8007 | 0.8965 | 7.2406 | **10.6514** | **0.7162** | 0.8602 | 0.5766 | 0.7815 | 0.4848 | 0.7884 | 0.6941 | 0.8645 | 0.2525 | 0.6509 |
| BRR (Zhan et al., 2025) | SF | 6.2294 | 8.8952 | 0.8013 | 0.8964 | 7.2067 | 10.6573 | 0.7159 | 0.8588 | 0.5909 | 0.8037 | 0.4554 | 0.7917 | 0.6862 | 0.8652 | 0.2504 | 0.6469 |
| VM | SF | 6.0859 | 8.6171 | 0.8134 | 0.9060 | 7.4787 | 11.9665 | 0.6415 | 0.8373 | 0.4999 | 0.6641 | 0.6276 | 0.8065 | 0.6949 | 0.8584 | 0.2631 | 0.5940 |
| AugSelfTr | SF | 6.1548 | 8.6981 | 0.8100 | 0.9031 | 7.4219 | 11.8506 | 0.6483 | 0.8415 | 0.5092 | 0.6825 | 0.6069 | 0.7975 | 0.7108 | 0.8669 | 0.2485 | 0.5884 |
| MERCI | SF | 5.9570 | 8.3558 | 0.8247 | 0.9104 | **7.0126** | 10.9240 | 0.7013 | 0.8590 | 0.4674 | 0.6347 | 0.6602 | 0.8230 | 0.6949 | 0.8917 | 0.2047 | 0.6105 |
| MERCI w. FN | SF | 5.8613 | 8.2914 | 0.8273 | 0.9116 | 7.0568 | 11.0425 | 0.6949 | 0.8552 | 0.4652 | 0.6338 | 0.6612 | 0.8241 | 0.6786 | 0.8647 | 0.2520 | 0.6151 |
| MERCI-R | SF | 5.8263 | 8.2325 | 0.8298 | 0.9117 | 7.0802 | 10.9199 | 0.7014 | 0.8594 | 0.4661 | 0.6340 | 0.6609 | 0.8238 | 0.6909 | 0.8841 | 0.2179 | 0.6070 |
| MERCI-R w. FN | SF | 5.8571 | 8.3965 | 0.8229 | 0.9094 | 7.1764 | 11.0313 | 0.6954 | **0.8604** | 0.4646 | 0.6290 | 0.6664 | 0.8259 | **0.6649** | **0.8503** | **0.2758** | 0.6159 |
| Oracle | - | 5.0135 | 7.0991 | 0.8758 | 0.9373 | 4.9198 | 7.2291 | 0.8690 | 0.9336 | 0.2794 | 0.3947 | 0.8686 | 0.9335 | 0.2729 | 0.4115 | 0.8307 | 0.9127 |

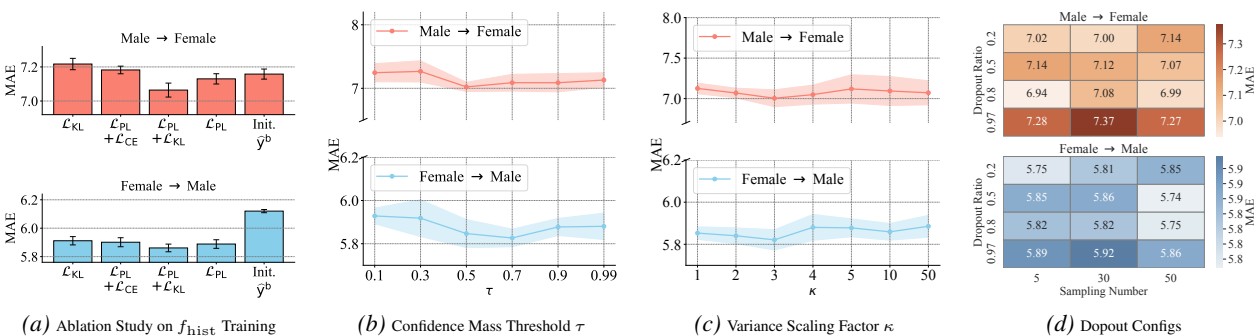

*(a)* Ablation Study on $f_{\text{hist}}$ Training   *(b)* Confidence Mass Threshold $\tau$   *(c)* Variance Scaling Factor $\kappa$   *(d)* Dopout Configs

*Figure 3. Ablation and hyperparameter sensitivity analysis on the UTKFace dataset*

for this task: VM and AugSelfTr (Zhang et al., 2022a). For our methods, **MERCI** and **MERCI-R** correspond to training with $\mathcal{L}_{\text{MERCI}}$ and $\mathcal{L}_{\text{MERCI-R}}$, respectively, while "w. FN" denotes the use of feature norm regularization ($\mathcal{L}_{\text{FN}}$).

Further details, including datasets and baseline descriptions, experimental configurations, and evaluation metrics, are provided in Appendix D.

### 5.1. Overall Results, Analysis and Discussion

**Overall Results.** Comprehensive experimental results on four datasets are presented in Tables 1, 2, E.2 and E.4. Overall, our proposed method, including all variants of MERCI, consistently improves the performance of the source model across all evaluation metrics without accessing any source data. In most cases, MERCI outperforms source-free (SF) and data-free (DF) methods; on certain tasks, it even surpasses UDA methods that have access to the original source data. Moreover, the performance gains are more obvious on more challenging tasks (i.e., those with higher source model MAE), demonstrating the reliability of our approach under challenging scenarios. Additional experiments validating the robustness of MERCI to varying distribution shift severities are presented in Appendix E.5.

**Ablation Study.** We first conduct ablation experiments to investigate the different training strategies for the **histogram**

head $f_{\text{hist}}$. As shown in Figure 3a, applying the partial label loss ($\mathcal{L}_{\text{PL}}$) improves regression performance, and combining it with the KL divergence loss ($\mathcal{L}_{\text{KL}}$) consistently yields the best results. To further assess the effectiveness of incorporating the histogram head $f_{\text{hist}}$, we conduct an ablation study where the regressor is directly trained using the histogram expectation estimated from the partial label set. This variant, denoted as Init. $\widehat{\mathbf{y}}^{\mathrm{b}}$ in Figure 3a, exhibits unstable performance. Additional analysis of the interaction between $f_{\text{hist}}$ and $f_{\text{reg}}$ is provided in Appendix E.9, highlighting the importance of a learnable histogram head. **For the regressor**, as shown in Tables 1 and 2, MERCI-R consistently outperforms MERCI by employing a robust RMSE-based loss that is more tolerant to noisy supervision. Additionally, feature scale normalization ($\mathcal{L}_{\text{FN}}$) contributes to stabilizing MSE-based training in more challenging SFDAR setting, such as on the Biwi-Kinect (Male → Female). Further ablation results and discussions are provided in Appendix E.2.

**Histogram Information.** To better assess the quality of the histogram and its role in correcting supervision signals, we summarize key statistics across all adaptation tasks in Table 3. The "*Histogram Information*" columns demonstrate that our self-adaptive histogram construction consistently produces reasonable bin widths and corresponding bin numbers, validating its applicability across datasets. To further understand the benefit of introducing a histogram head $f_{\text{hist}}$

*Table 3. Summary of histogram information across four datasets and different adaptation tasks.*

| Dataset | Task | Histogram Information | | Partial Label Set Quality | | Continuous Pseudo-Label Quality | | |
| | | Number of Bins ($K$) | Bin Length | Correct Label Coverage | Top-1 pseudo-label Accuracy | Source Model MAE | Init. $\bar{y}$ MAE | Best $\bar{y}$ MAE |
|---|---|---|---|---|---|---|---|---|
| UTKFace | Female → Male | 40.40 | 3.26 | 0.85 | 0.21 | 6.55 | 6.39 | 5.77 |
| UTKFace | Male → Female | 65.00 | 1.93 | 0.75 | 0.15 | 7.94 | 7.63 | 6.94 |
| Biwi-Kinect | Female → Male (pitch) | 36.20 | 3.79 | 0.87 | 0.24 | 5.17 | 4.61 | 4.53 |
| Biwi-Kinect | Female → Male (yaw) | 40.20 | 3.85 | 0.99 | 0.36 | 3.60 | 3.66 | 3.39 |
| Biwi-Kinect | Female → Male (roll) | 95.60 | 0.85 | 0.68 | 0.07 | 5.07 | 5.02 | 4.87 |
| Biwi-Kinect | Male → Female (pitch) | 36.00 | 3.88 | 0.69 | 0.14 | 8.70 | 8.74 | 7.40 |
| Biwi-Kinect | Male → Female (yaw) | 36.20 | 3.90 | 0.83 | 0.23 | 6.33 | 6.56 | 5.16 |
| Biwi-Kinect | Male → Female (roll) | 64.20 | 1.11 | 0.67 | 0.09 | 5.34 | 5.30 | 5.07 |
| Digits | SVHN → MNIST | 20.20 | 0.80 | 0.75 | 0.25 | 1.72 | 1.77 | 1.51 |
| Digits | MNIST → SVHN | 40.80 | 0.56 | 0.54 | 0.11 | 2.91 | 3.08 | 2.49 |
| California House | Near Bay → Far Bay | 74.00 | 0.26 | 0.97 | 0.13 | 0.73 | 0.71 | 0.66 |
| California House | Far Bay → Near Bay | 66.00 | 0.17 | 0.98 | 0.12 | 0.52 | 0.48 | 0.47 |

and the partial label loss, we evaluate the "*Partial Label Set Quality*" and find that ground-truth regression values are well covered by the assigned intervals, even though the top-1 pseudo-label is unreliable. This highlights that the partial label set as a whole could provide meaningful and robust supervision for label correction under weak supervision. Finally, we compare the initial and best "*Continuous Pseudo-Labels*" $\bar{y}$, taking the source model prediction as reference. The initial pseudo-labels (Init. $\bar{y}$) outperform the source model slightly but exhibit instability, whereas the best pseudo-labels (Best $\bar{y}$) obtained during adaptation are markedly more accurate, often matching or surpassing the final adapted regressor. These results highlight the effectiveness of bi-directional learning between the regression and histogram heads, and justify the introduction of $f_{\mathrm{hist}}$ instead of directly supervising the regressor with the initial histogram information.

**Hyperparameter Sensitivity.** We perform a sensitivity analysis of four key hyperparameters: the confidence mass threshold $\tau$ in $\mathcal{L}_{\mathrm{MSE}}$ and $\mathcal{L}_{\mathrm{RMSE}}$, the variance scaling factor $\kappa$ in $\mathcal{L}_{\mathrm{KL}}$, and the dropout configurations—namely the dropout ratio $\rho$ and the sampling number $M$. As shown in Figures 3b and 3c, MERCI maintains stable performance over a wide range of $\tau$ (0.3–0.7) and $\kappa$ (1–4). We thus fix $\tau = 0.68$ and $\kappa = 3$ in all experiments. Since dropout configurations affect the initial histogram construction, we observe stable performance under moderate dropout ratios and sampling numbers, and accordingly set $\rho = 0.8$ and $M = 30$ for all datasets. For the remaining hyperparameters, we set the moving average coefficient $\alpha$ to 0.5 across all datasets, and treat the coefficients of different loss terms (e.g., $\lambda_{\mathrm{PL}}$, $\lambda_{\mathrm{prior}}$, $\lambda_{\mathrm{RMSE}}$) as learnable parameters to simplify the selection process and better fit the SFDAR problem. Additional sensitivity analysis is provided in Appendix E.3.

**Feature Representation Learning.** We present UMAP visualizations (McInnes et al., 2018) of the target feature space before and after adaptation on Biwi-Kinect and UTK-Face datasets. As shown in Figures 4 and E.6, source

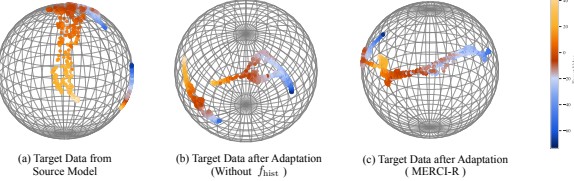

(a) Target Data from Source Model   (b) Target Data after Adaptation (Without $f_{\mathrm{hist}}$)   (c) Target Data after Adaptation (MERCI-R)

*Figure 4. UMAP visualization of target data feature space on Biwi-Kinect dataset*

features retain partial ordering but remain loosely structured, whereas adapted features are more locally compact. Moreover, incorporating $f_{\mathrm{hist}}$ introduces global diversity. These observations are consistent with the criteria of a well-performing regressor (Zhang et al., 2023) and further support the effectiveness of the proposed discretized information and the histogram head. Detailed analyses are provided in Appendix E.7.

Additional experimental results, including calibration analysis between the source and target domains as well as computational efficiency evaluations, are provided in Appendices E.10 and E.11.

# 6. Conclusion

The conditional distribution of the target label is a vital yet often overlooked aspect in deep regression learning, especially in SFDAR, where obtaining such information becomes particularly challenging. In this work, we propose MERCI, a novel framework tailored for SFDAR. By leveraging a learned, sample-wise histogram to approximate the discretized conditional density of target labels, MERCI effectively captures the underlying distributional characteristics of target data without requiring ground-truth labels or source data. It further generates uncertainty-aware pseudo-labels through truncated histogram expectations to facilitate robust regressor adaptation. Supported by both theoretical analysis and comprehensive experiments, MERCI demonstrates strong and consistent performance across a range of regression tasks under domain shift. This work provides an

insightful direction for modeling and leveraging uncertainty in regression under domain shift, paving the way for more reliable and generalizable adaptive regression framework.

## Acknowledgements

Gezheng Xu, Qiuhao Zeng, Charles Ling and Boyu Wang are supported by the Natural Sciences and Engineering Research Council of Canada (NSERC) Discovery Grants program. Qi Chen is supported by research funding from the ELLIS Institute Finland and the Department of Computer Science at Aalto University.

## Impact Statement

This paper presents work whose goal is to advance the field of Machine Learning. There are many potential societal consequences of our work, none which we feel must be specifically highlighted here.

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

# A. Use of Large Language Models (LLMs)

We acknowledge the use of LLMs in this work solely as a general-purpose assistant tool, primarily for grammar checking, sentence polishing, and writing refinement, which did not contribute to the conception, methodology, analysis, or conclusions of this work. All substantive contributions of this submission, spanning problem formulation, idea and methodology development, theoretical analysis, experimental design and implementation, result interpretation, and manuscript preparation, are entirely attributed to the human authors.

# B. Technical Details

Before delving into the technical details, we present a notation table that summarizes the key symbols used throughout the paper. For each entry, we provide the notation in the first column, its description in the second column, and the section where it first appears in the third column.

*Table B.1. Notation used throughout the paper.*

| Notations | Descriptions | First appearance |
|---|---|---|
| X/Y/Z | Random variables of input/label/feature representation | Section 1 |
| $\mathcal{H}(\cdot\|\cdot)$ | Conditional Entropy | Section 1 |
| $\mathcal{X} \subset \mathbb{R}^d$ | Input space | Section 3 |
| $\mathcal{Y} \subset \mathbb{R}$ | Label space | Section 3 |
| $P^{\mathrm{S}}_{\mathrm{xy}}$; $\mathcal{D}_{\mathrm{S}}$ | Underlying distribution over $\mathcal{X} \times \mathcal{Y}$ related to source domain; Unavailable source domain data $\mathcal{D}_{\mathrm{S}} \triangleq \{\mathrm{x}^{\mathrm{s}}_i, \mathrm{y}^{\mathrm{s}}_i\}^{N_{\mathrm{S}}}_{i=1}$ | Section 3 |
| $P^{\mathrm{T}}_{\mathrm{xy}}$; $\mathcal{D}_{\mathrm{T}}$ | Underlying distribution over $\mathcal{X} \times \mathcal{Y}$ related to target domain; Unlabeled target domain data $\mathcal{D}_{\mathrm{T}} \triangleq \{\mathrm{x}^{\mathrm{T}}_i\}^{N_{\mathrm{T}}}_{i=1}$ | Section 3 |
| $g_{\mathrm{S}}/g_{\mathrm{T}}/g$ | Feature extractors of source/target/general model | Section 3 |
| $f_{\mathrm{reg,S}}/f_{\mathrm{reg,T}}/f_{\mathrm{reg}}$ | Regression heads of source/target/general model | Section 3 |
| $f_{\mathrm{hist}}$ | Histogram head of MERCI model | Section 4 |
| $K$ | Number of histogram bins | Section 3 |
| $\mu, \sigma, \kappa$ | mean, standard deviation, and variance scaling factor of Gaussian Prior | Section 4 |
| $\bar{\mathrm{y}}_i$ | Continuous Pseudo-Label | Section 4 |
| $\widetilde{\mathrm{y}}_k$ | Representative regression value of the $k$-th histogram bin | Section 4.1 |
| $\widetilde{p}_i \in \Delta^{K-1}$ | Empirical discrete distribution over $K$ histogram bins of $\mathrm{x}^{\mathrm{T}}_i$ | Section 4.1 |
| $\widetilde{\pi}$ | Discrete Prior Distribution | Section 4.1 |
| $\widehat{\mathrm{y}}_i$ | Uncertainty-aware regression prediction (continuous) | Section 4.1 |
| $\widehat{\mathrm{y}}^{\mathrm{b}}_i, \widehat{\mathbf{y}}^{\mathrm{b}}_i$ | Pseudo-histogram label (i.e., bin index) and partial bin index set for $\mathrm{x}^{\mathrm{T}}_i$ | Section 4.1 |
| $\widetilde{p}_i$ | Empirical distribution over the pseudo-label-feature pairs $\{(\mathrm{z}^{\mathrm{T}}_i, \bar{\mathrm{y}}_i)\}^{N_{\mathrm{T}}}_{i=1}$ | Section 4.1 |
| $\ell, \mathcal{L}$ | Instance-level and mini-batch/dataset level loss term | Section 4.1 |
| $\mathcal{H} = \{(\mathrm{x}^{\mathrm{T}}_i, \widetilde{p}^{\mathrm{old}}_i, \widehat{\mathbf{y}}^{\mathrm{b}}_i)\}^{N_{\mathrm{T}}}_{i=1}$ | Histogram information set | Section C.2 |
| $\texttt{DROPOUT}(\cdot, \rho)$ | Dropout function with ratio $\rho \in [0, 1]$ | Section C.2 |

## B.1. Preliminaries

Let $\mathcal{S}$ denote a set that admits Polish topology, let $\mathcal{P}^{\dagger}(\mathcal{S})$ denote the set of finitely-additive measures over $\mathcal{S}$, let $\mathcal{P}(\mathcal{S}) \subseteq \mathcal{P}^{\dagger}(\mathcal{S})$ stand for the set of Borel probability measures supported on $\mathcal{S}$, and let $\mathcal{F}_b(\mathcal{S})$ represent the set of all bounded and measurable functions mapping from $\mathcal{S}$ to $\mathbb{R}$

We first present several useful definitions and lemmas following Husain & Knoblauch (2022).

**Definition B.1** (Divergences)**.** A divergence is a function $\mathrm{d} : \mathcal{P}(\mathcal{S}) \times \mathcal{P}(\mathcal{S}) \to \mathbb{R}$, such that for any $\mu_1, \mu_2 \in \mathcal{P}(\mathcal{S})$, (1) $\mathrm{d}(\mu_1, \mu_2) \geq 0$; (2) $\mathrm{d}(\mu_1, \mu_2) = 0 \Leftrightarrow \mu_1 = \mu_2$; and (3) $\mathrm{d}$ is proper convex lower semi-continuous in its first argument.

**Definition B.2** (Optimal transport divergence; Blanchet et al. (2019))**.** Let $c : \mathbb{R}^q \times \mathbb{R}^q \to [0, +\infty]$ be any lower semi-

continuous function such that $c(u, u) = 0$ for every $u \in \mathbb{R}^q$. Given two probability distributions $P_1(\cdot)$ and $P_2(\cdot)$ supported on $\mathbb{R}^q$, the optimal transport divergence between $P_1$ and $P_2$, denoted by $\mathsf{W}_c(P_1, P_2)$, is defined as

$$\mathsf{W}_c(P_1, P_2) = \inf\{\mathbb{E}_\pi\{c(U_1, U_2)\} : \pi \in \mathrm{Cpl}(P_1, P_2)\},$$

where $\mathrm{Cpl}(P_1, P_2)$, sometimes called the coupling set of $P_1$ and $P_2$, comprises all probability measures on the product space $\mathbb{R}^q \times \mathbb{R}^q$ such that their marginal measures are $P_1(\cdot)$ and $P_2(\cdot)$. In particular, when taking the square cost function $c(u_1, u_2) = \|u_1 - u_2\|_2^2$ for $u_1, u_2 \in \mathbb{R}^q$, we denote $\mathsf{W}_c$ as $\mathsf{W}_2$.

**Definition B.3** (Legendre-Fenchel conjugate). For a given $\pi \in \mathcal{P}(\mathcal{S})$, the Legendre-Fenchel conjugate of $\mathsf{d}(\cdot, \pi) : \mathcal{P}(\mathcal{S}) \to \mathbb{R}$ is defined as below:

$$\mathsf{d}_\pi^*(\ell) = \sup_{\mu \in \mathcal{P}^\dagger(\mathcal{S})} \left\{ \int_\mathcal{S} \ell \, d\mu - \mathsf{d}(\mu, \pi) \right\} \text{ for all } \ell \in \mathcal{F}_b(\mathcal{S}).$$

**Definition B.4** (Closed convex hull). For a set $\Pi$, the *convex hull* of $\Pi$, denoted $\mathrm{co}(\Pi)$, is the smallest convex set containing $\Pi$. Equivalently, it can be defined as:

$$\mathrm{co}(\Pi) = \left\{ \sum_{i=1}^n \lambda_i \mu_i : \mu_i \in \Pi, \lambda_i \geq 0, \sum_{i=1}^n \lambda_i = 1, n \in \mathbb{N} \right\}.$$

The *closed convex hull* of $\Pi$, denoted $\overline{\mathrm{co}}(\Pi)$, is the smallest closed convex set that includes $\Pi$, and can be equivalently defined as the closure of $\mathrm{co}(\Pi)$.

*Remark* B.5. For ease of presentation, we assume in Equation (1) (Section 4.1) that $\Pi$—the set of distributions over $\widetilde{\mathcal{Y}}$ induced by the classification model—is convex and closed. If this assumption does not hold, we may instead work with its closed convex hull $\overline{\mathrm{co}}(\Pi)$ in (1).

**Lemma B.6.** *For any $\mu, \pi \in \mathcal{P}(\mathcal{S})$, the following holds:*

$$\mathsf{d}(\mu, \pi) = \sup_{\ell \in \mathcal{F}_b(\mathcal{S})} \left\{ \mathbb{E}_\mu(\ell) - \mathsf{d}_\pi^*(\ell) \right\}.$$

**Lemma B.7.** *Let $\ell : \mathcal{P}(\mathcal{S}) \to \mathbb{R}$ denote a convex and lower semicontinuous function. For any $\pi \in \mathcal{P}(\mathcal{S})$, divergence $\mathsf{d}$, $\lambda > 0$, and set $\Pi \subseteq \mathcal{P}(\mathcal{S})$, define a function $F : \mathcal{P}(\mathcal{S}) \times \mathcal{F}_b(\mathcal{S}) \to \mathbb{R}$ as*

$$F(p, \ell) = \ell(p) + \lambda \left\{ \mathbb{E}_p(\ell) - \mathsf{d}_\pi^*(\ell) \right\} + \iota_{\overline{co}(\Pi)}(p),$$

*where $\iota_{\overline{co}(\Pi)}(p) = +\infty$ if $p \notin \overline{co}(\Pi)$, and $0$ otherwise. Then, it holds that*

$$\inf_{p \in \mathcal{P}(\mathcal{S})} \sup_{\ell \in \mathcal{F}_b(\mathcal{S})} F(p, \ell) = \sup_{\ell \in \mathcal{F}_b(\mathcal{S})} \inf_{p \in \mathcal{P}(\mathcal{S})} F(p, \ell).$$

**Lemma B.8** (Fenchel-Young inequality). *Let $\mathcal{S}$ be a complete normed vector space and let $\mathcal{S}^*$ be the dual space to $\mathcal{S}$. For a function $f : \mathcal{S} \to \mathbb{R} \cup \{-\infty, +\infty\}$, its convex conjugate is the function $f^* : \mathcal{S}^* \to \mathbb{R} \cup \{-\infty, +\infty\}$ defined as: $f^*(x^*) = \sup \{\langle x^*, x \rangle - f(x) : x \in \mathcal{S}\}$ for any $x^* \in \mathcal{S}^*$. Then, it holds for every $x \in \mathcal{S}$ and $x^* \in \mathcal{S}^*$:*

$$\langle x^*, x \rangle \leq f(x) + f^*(x^*).$$

## B.2. Proof of Proposition 4.1

We now present the proof of Proposition 4.1 by modifying the proofs in Husain & Knoblauch (2022).

*Proof.* By definition, we have

$$\ell_{\mathrm{PL}}(\widetilde{p}^\star; \widehat{\mathbf{y}}^b) + \mathbb{E}_{\widetilde{p}^\star} \left\{ \ell^\star(\mathrm{Y}^\mathrm{T}) \right\} \geq \inf_{\widetilde{p} \in \Pi} \left[ \ell_{\mathrm{PL}}(\widetilde{p}; \widehat{\mathbf{y}}^b) + \mathbb{E}_{\widetilde{p}} \left\{ \ell^\star(\mathrm{Y}^\mathrm{T}) \right\} \right]. \tag{B1}$$

We now prove the other direction in the following.

The optimization problem in (1) can be expressed as below:

$$
\inf_{\widetilde{p} \in \Pi} \{\ell_{\mathrm{PL}}(\widetilde{p}; \widehat{\mathbf{y}}^{\mathrm{b}}) + \lambda_{\mathrm{prior}} \mathrm{d}(\widetilde{p}, \widetilde{\pi})\}
$$

$$
= \inf_{\widetilde{p} \in \Pi} \left[ \ell_{\mathrm{PL}}(\widetilde{p}; \widehat{\mathbf{y}}^{\mathrm{b}}) + \lambda_{\mathrm{prior}} \sup_{\ell \in \mathcal{F}_b(\mathcal{S})} \{\mathbb{E}_{\widetilde{p}}(\ell) - \mathrm{d}_{\widetilde{\pi}}^*(\ell)\} \right]
$$

$$
= \inf_{\widetilde{p} \in \mathcal{P}(\widetilde{\mathcal{Y}})} \sup_{\ell \in \mathcal{F}_b(\mathcal{S})} \left[ \ell_{\mathrm{PL}}(\widetilde{p}; \widehat{\mathbf{y}}^{\mathrm{b}}) + \lambda_{\mathrm{prior}} \{\mathbb{E}_{\widetilde{p}}(\ell) - \mathrm{d}_{\widetilde{\pi}}^*(\ell)\} + \iota_{\Pi}(\widetilde{p}) \right]
$$

$$
= \sup_{\ell \in \mathcal{F}_b(\mathcal{S})} \inf_{\widetilde{p} \in \mathcal{P}(\widetilde{\mathcal{Y}})} \left[ \ell_{\mathrm{PL}}(\widetilde{p}; \widehat{\mathbf{y}}^{\mathrm{b}}) + \lambda_{\mathrm{prior}} \{\mathbb{E}_{\widetilde{p}}(\ell) - \mathrm{d}_{\widetilde{\pi}}^*(\ell)\} + \iota_{\Pi}(\widetilde{p}) \right]
$$

$$
= \sup_{\ell \in \mathcal{F}_b(\mathcal{S})} \left[ \inf_{\widetilde{p} \in \mathcal{P}(\widetilde{\mathcal{Y}})} \{\ell_{\mathrm{PL}}(\widetilde{p}; \widehat{\mathbf{y}}^{\mathrm{b}}) + \lambda_{\mathrm{prior}} \mathbb{E}_{\widetilde{p}}(\ell) + \iota_{\Pi}(\widetilde{p})\} - \lambda_{\mathrm{prior}} \mathrm{d}_{\widetilde{\pi}}^*(\ell) \right]
$$

$$
= \sup_{\ell \in \mathcal{F}_b(\mathcal{S})} \left[ \inf_{\widetilde{p} \in \Pi} \{\ell_{\mathrm{PL}}(\widetilde{p}; \widehat{\mathbf{y}}^{\mathrm{b}}) + \lambda_{\mathrm{prior}} \mathbb{E}_{\widetilde{p}}(\ell)\} - \lambda_{\mathrm{prior}} \mathrm{d}_{\widetilde{\pi}}^*(\ell) \right]
$$

$$
= \sup_{\ell' \in \mathcal{F}_b(\mathcal{S})} \left[ \inf_{\widetilde{p} \in \Pi} \{\ell_{\mathrm{PL}}(\widetilde{p}; \widehat{\mathbf{y}}^{\mathrm{b}}) + \mathbb{E}_{\widetilde{p}}(\ell')\} - \lambda_{\mathrm{prior}} \mathrm{d}_{\widetilde{\pi}}^*\left(\frac{\ell'}{\lambda_{\mathrm{prior}}}\right) \right] \tag{B2}
$$

where the first step follows from Lemma B.6, the third step comes from Lemma B.7 and the assumption that $\Pi$ is closed and convex, and the last step holds by letting $\ell' = \lambda_{\mathrm{prior}} \ell$ and applying the definition of $\mathcal{F}_b(\mathcal{S})$.

Consequently, we further obtain that

$$
\inf_{\widetilde{p} \in \Pi} [\ell_{\mathrm{PL}}(\widetilde{p}; \widehat{\mathbf{y}}^{\mathrm{b}}) + \mathbb{E}_{\widetilde{p}}\{\ell^{\star}(\mathrm{Y}^{\mathrm{T}})\}] - [\ell_{\mathrm{PL}}(\widetilde{p}^{\star}; \widehat{\mathbf{y}}^{\mathrm{b}}) + \mathbb{E}_{\widetilde{p}^{\star}}\{\ell^{\star}(\mathrm{Y}^{\mathrm{T}})\}]
$$

$$
= \inf_{\widetilde{p} \in \Pi} [\ell_{\mathrm{PL}}(\widetilde{p}; \widehat{\mathbf{y}}^{\mathrm{b}}) + \mathbb{E}_{\widetilde{p}}\{\ell^{\star}(\mathrm{Y}^{\mathrm{T}})\}] - \lambda_{\mathrm{prior}} \mathrm{d}_{\widetilde{\pi}}^*\left(\frac{\ell^{\star}}{\lambda_{\mathrm{prior}}}\right)
$$

$$
+ \lambda_{\mathrm{prior}} \mathrm{d}_{\widetilde{\pi}}^*\left(\frac{\ell^{\star}}{\lambda_{\mathrm{prior}}}\right) - [\ell_{\mathrm{PL}}(\widetilde{p}^{\star}; \widehat{\mathbf{y}}^{\mathrm{b}}) + \mathbb{E}_{\widetilde{p}^{\star}}\{\ell^{\star}(\mathrm{Y}^{\mathrm{T}})\}]
$$

$$
= \sup_{\ell' \in \mathcal{F}_b(\mathcal{S})} \left\{ \inf_{\widetilde{p} \in \Pi} [\ell_{\mathrm{PL}}(\widetilde{p}; \widehat{\mathbf{y}}^{\mathrm{b}}) + \mathbb{E}_{\widetilde{p}}\{\ell'(\mathrm{Y}^{\mathrm{T}})\}] - \lambda_{\mathrm{prior}} \mathrm{d}_{\widetilde{\pi}}^*\left(\frac{\ell'}{\lambda_{\mathrm{prior}}}\right) \right\}
$$

$$
+ \lambda_{\mathrm{prior}} \mathrm{d}_{\widetilde{\pi}}^*\left(\frac{\ell^{\star}}{\lambda_{\mathrm{prior}}}\right) - [\ell_{\mathrm{PL}}(\widetilde{p}^{\star}; \widehat{\mathbf{y}}^{\mathrm{b}}) + \mathbb{E}_{\widetilde{p}^{\star}}\{\ell^{\star}(\mathrm{Y}^{\mathrm{T}})\}]
$$

$$
= \inf_{\widetilde{p} \in \Pi} \{\ell_{\mathrm{PL}}(\widetilde{p}; \widehat{\mathbf{y}}^{\mathrm{b}}) + \lambda_{\mathrm{prior}} \mathrm{d}(\widetilde{p}, \widetilde{\pi})\} + \lambda_{\mathrm{prior}} \mathrm{d}_{\widetilde{\pi}}^*\left(\frac{\ell^{\star}}{\lambda_{\mathrm{prior}}}\right) - [\ell_{\mathrm{PL}}(\widetilde{p}^{\star}; \widehat{\mathbf{y}}^{\mathrm{b}}) + \mathbb{E}_{\widetilde{p}^{\star}}\{\ell^{\star}(\mathrm{Y}^{\mathrm{T}})\}]
$$

$$
= \ell_{\mathrm{PL}}(\widetilde{p}^{\star}; \widehat{\mathbf{y}}^{\mathrm{b}}) + \lambda_{\mathrm{prior}} \mathrm{d}(\widetilde{p}^{\star}, \widetilde{\pi}) + \lambda_{\mathrm{prior}} \mathrm{d}_{\widetilde{\pi}}^*\left(\frac{\ell^{\star}}{\lambda_{\mathrm{prior}}}\right) - [\ell_{\mathrm{PL}}(\widetilde{p}^{\star}; \widehat{\mathbf{y}}^{\mathrm{b}}) + \mathbb{E}_{\widetilde{p}^{\star}}\{\ell^{\star}(\mathrm{Y}^{\mathrm{T}})\}]
$$

$$
= \lambda_{\mathrm{prior}} \left\{ \mathrm{d}(\widetilde{p}^{\star}, \widetilde{\pi}) + \mathrm{d}_{\widetilde{\pi}}^*\left(\frac{\ell^{\star}}{\lambda_{\mathrm{prior}}}\right) - \mathbb{E}_{\widetilde{p}^{\star}}\left(\frac{\ell^{\star}}{\lambda_{\mathrm{prior}}}\right) \right\}
$$

$$
\geq 0, \tag{B3}
$$

where the second step is due to the definition of $\ell^{\star}$ given in Proposition 4.1, the third step follows from (B2), the fourth step comes from the definition of $p_{\mathrm{hist}}^{\star}$, and the last step comes from the Fenchel-Young inequality in Lemma B.8. The proof is established by combining (B1) and (B3). $\qquad\square$

*Remark* B.9. Let $\Theta$ denote the sample space, $\pi$ a prior on $\Theta$, and $p_{\theta}$ the likelihood model indexed by $\theta \in \Theta$. Given prior $\pi$, observed data $x$, and a parameterized subset $\Pi \subset \mathcal{P}(\Theta)$, the associated variational posterior (Blei et al., 2017) is defined as

$$
q_{\mathrm{VI}} \in \operatorname*{arg\,inf}_{q \in \Pi} \left[ \mathbb{E}_{q(\theta)}\{-\log p_{\theta}(x)\} + \mathrm{d}_{\mathrm{KL}}(q, \pi) \right].
$$

By extending the negative log-likelihood function to a general loss function $\ell \in \mathcal{F}_b(\Theta)$ and generalizing $\mathrm{d}_{\mathrm{KL}}$ to any divergence $\mathrm{d}$ as defined in Definition B.1, the generalized variational posterior with $\lambda > 0$ (Husain & Knoblauch, 2022) is

defined as

$$q_{\mathrm{GVI}} \in \arg\inf_{q \in \Pi} \left\{ \mathbb{E}_{q(\theta)}\{\ell(\theta; x)\} + \lambda \mathtt{d}(q, \pi) \right\}.$$

Similarly, the optimization problem (1) can be expressed as:

$$q^\star \in \arg\inf_{q \in \Pi} \left\{ \ell(q; x) + \lambda \mathtt{d}(q, \pi) \right\},$$

which can be viewed as a generalization of $q_{\mathrm{GVI}}$. Here, the loss function $\ell$ need not depend linearly on $q$; we only require that the mapping $q \mapsto \ell(q; x)$ is convex and lower semicontinuous.

## C. Algorithm

### C.1. Implementation Details for Histogram Approximation

In this section, we elaborate on the implementation details of the histogram distribution approximation and summarize the complete training objective.

**Self-Adaptive Histogram Construction.** To discretize the continuous regression outputs into histogram bins, it is crucial to choose an appropriate bin width. On one hand, excessively wide bins may cause a large portion of the data to concentrate within a few intervals, leading to degraded learning performance or training collapse. On the other hand, overly narrow bins may result in sparse class distributions, making the associated classification task difficult and reducing predictive accuracy. To balance these two extremes, we introduce a *self-adaptive histogram binning method* that automatically adjusts the bin structure based on the distribution of the regression head's outputs.

Specifically, for each input $\mathrm{x}_i^{\mathrm{T}}$ from the target dataset $\mathcal{D}_{\mathrm{T}}$, let $\widehat{\mathbf{y}}_i \triangleq \{\widehat{y}_{i,1}, \ldots, \widehat{y}_{i,M}\}$ denote a collection of predictions from the regression head, obtained through stochastic inference techniques such as dropout. Let $\widehat{\mathcal{Y}} \triangleq \{\widehat{\mathbf{y}}_i\}_{i=1}^{N_{\mathrm{T}}}$ denote the aggregated set of all such predictions. Then, we apply Gaussian kernel density estimation (KDE) to $\widehat{\mathcal{Y}}$, and denote the resulting density function as $\widehat{h}(\cdot)$. To estimate the central tendency of the target label distribution, we identify the peak (mode) of the estimated density as: $y_{\mathrm{peak}} \in \arg\max_{y \in \mathcal{Y}} \widehat{h}(y)$, and let $\alpha_{\mathrm{peak}}$ denote the corresponding quantile level of $y_{\mathrm{peak}}$ in the cumulative distribution.

The *bin width* is then defined adaptively as:

$$\mathtt{b} \triangleq Q_{\alpha_{\mathrm{peak}}+0.05}(\widehat{h}) - Q_{\alpha_{\mathrm{peak}}-0.05}(\widehat{h})$$

where $Q_\alpha(\widehat{h})$ denotes the $\alpha$-quantile of the density function $\widehat{h}$ for $\alpha \in [0, 1]$. This design uses the most concentrated 10% of the data to guide the choice of bin width, allowing the binning scheme to adapt to the local density structure of the pproximated target label distribution. Given the overall range of the regression outputs and the adaptive bin width, we compute the total *number of bins* as:

$$K \triangleq \mathtt{int}\left\{ \frac{\max(\widehat{\mathcal{Y}}) - \min(\widehat{\mathcal{Y}})}{\mathtt{b}} \right\}. \tag{C1}$$

Finally, we assign a *representative value* $\widetilde{y}_k$ for each bin $\mathcal{Y}_k$ in the partition $\mathcal{Y} = \cup_{k=1}^K \mathcal{Y}_k$, we assign a *representative value* $\widetilde{y}_k$, which is determined by taking into account the overall skewness of the estimated target distribution. Specifically, let $\eta$ be a small tolerance parameter. Define $\mathcal{Y}_{k,\,\mathrm{left}}$, $\mathcal{Y}_{k,\,\mathrm{mid}}$, and $\mathcal{Y}_{k,\,\mathrm{right}}$ denote the left endpoint, midpoint, and right endpoint of the $k$th bin, respectively. Then, the representative value $\widetilde{y}_k$ is set as follows:

$$\widetilde{y}_k \triangleq \begin{cases} \mathcal{Y}_{k,\,\mathrm{left}}, & \text{if } \alpha_{\mathrm{peak}} < 0.5 - \eta \quad \text{(left-skewed)} \\[2mm] \mathcal{Y}_{k,\,\mathrm{right}}, & \text{if } \alpha_{\mathrm{peak}} > 0.5 + \eta \quad \text{(right-skewed)} \\[2mm] \mathcal{Y}_{k,\,\mathrm{mid}}, & \text{otherwise} \quad \text{(approximately symmetric)} \end{cases} \tag{C2}$$

In our implementation, we set $\eta = 0.02$.

**Partial Label Loss.** After adaptively partitioning the regression label space into $K$ bins, each continuous prediction $\widehat{y}_{i,j} \in \widehat{\mathbf{y}}_i$ (for $j \in [M]$) is mapped to a bin index $\widehat{y}_{i,j}^{\mathrm{b}}$. We denote the resulting discrete labels as $\widehat{\mathbf{y}}_i^{\mathrm{b}} \triangleq \{\widehat{y}_{i,1}^{\mathrm{b}}, \ldots, \widehat{y}_{i,M}^{\mathrm{b}}\}$. Given the partial label set $\widehat{\mathbf{y}}_i^{\mathrm{b}} \subset [K]$ and the approximated histogram distribution $\widetilde{p}_i = h_{\mathrm{hist}}(\mathrm{x}_i^{\mathrm{T}}) \in \Delta^{K-1}$, we define the following partial label loss:

$$\ell_{\mathrm{PL}}(\widetilde{p}_i; \widehat{\mathbf{y}}_i^{\mathrm{b}}) \triangleq -\frac{1}{|\widehat{\mathbf{y}}_i^{\mathrm{b}}|} \sum_{k \in \widehat{\mathbf{y}}_i^{\mathrm{b}}} \log \widetilde{p}_i[k]. \tag{C3}$$

Then, with the partial label loss defined in (C3), for a batch $\mathcal{B}$, the batch-wise partial bin set loss is:

$$\mathcal{L}_{\mathrm{PL}} = \frac{1}{|\mathcal{B}|} \sum_{i \in \mathcal{B}} \ell_{\mathrm{PL}}(\widetilde{p}_i; \widehat{\mathbf{y}}_i^{\mathrm{b}}). \tag{C4}$$

Additionally, to further accommodate the source-free setting, preserve the continuity of the regression space, and retain the uncertainty information, we also introduce a method to refine the partial label set $\widehat{\mathbf{y}}_i^{\mathrm{b}}$ in a structured way. Specifically, we identify the index range spanned by $\widehat{\mathbf{y}}_i^{\mathrm{b}}$ as $k_{\min} \triangleq \min \widehat{\mathbf{y}}_i^{\mathrm{b}}$ and $k_{\max} \triangleq \max \widehat{\mathbf{y}}_i^{\mathrm{b}}$, and treat the interval $[k_{\min}, k_{\max}]$ as the core region of label predictions for $\mathrm{x}_i^{\mathrm{T}}$. To further reflect predictive uncertainty and promote smoother label assignment, we expand this interval by a relaxation factor proportional to the total number of bins. The resulting partial label (bin) set is then defined as:

$$\widehat{\mathbf{y}}_i^{\mathrm{b+}} \triangleq \{k \in [K] : k \in [k_{\min} - \lfloor \epsilon \cdot K \rfloor, k_{\min} + \lfloor \epsilon \cdot K \rfloor]\}.$$

where $\epsilon \in (0, 1)$ controls the extent of relaxation. This relaxed binning strategy allows the model to maintain a contiguous set of plausible labels, mitigating overconfidence and better accommodating the inherent uncertainty in the predictions. In our experiments, we set $\epsilon = 0.05$.

**Unimodal Prior.** For each $\mathrm{x}_i^{\mathrm{T}}$, we employ a unimodal Gaussian prior $\pi(\mathrm{y}|\mathrm{x}_i^{\mathrm{T}}) \sim \mathcal{N}(\mu_i, \sigma_i^2)$ for its label distribution, where the mean $\mu_i$ and variance $\sigma_i^2$ are constructed based on the histogram distribution obtained from the previous training iteration and the partial label set $\widehat{\mathbf{y}}_i^{\mathrm{b}}$.

Specifically, we first determine the bin index with the highest probability in the previous iteration's histogram $\widetilde{p}_i^{\mathrm{old}}$ (or initialize it as the empirical distribution over $\widehat{\mathbf{y}}_i^{\mathrm{b}}$), and denote this index as $k_\mu \triangleq \arg\max_{k \in [K]} \widetilde{p}_i^{\mathrm{old}}[k]$. The corresponding representative value in $\widetilde{\mathcal{Y}}$ is then selected as the mean: $\mu_i = \widetilde{y}_{k_\mu}$. To define the variance $\sigma_i^2$, we account for the spread of the refined label set around the mode. In particular, we set $\kappa\sigma_i = \mathtt{b} \cdot \max(|k_\mu - \min \widehat{\mathbf{y}}_i^{\mathrm{b}}|, |k_\mu - \max \widehat{\mathbf{y}}_i^{\mathrm{b}}|)$, where $\mathtt{b}$ is the bin width and $\kappa > 0$ is a tunable variance scaling factor. This formulation ensures that the prior variance adapts to the dispersion of the discrete label distribution.

Then, the discrete prior distribution $\widetilde{\pi}_i$ on $\widetilde{\mathcal{Y}}$ is obtained by approximating the probability mass of $\pi$ over the $K$ bins:

$$\widetilde{\pi}_i[k] = \frac{1}{\mathtt{C}_i} \cdot \mathtt{b} \, \varphi(\widetilde{y}_k; \mu_i, \sigma_i^2) \ \text{ for } k \in [K],$$

where $\varphi(\cdot; \mu_i, \sigma_i^2)$ denote the density function for the Gaussian distribution $\mathcal{N}(\mu_i, \sigma_i^2)$, and $\mathtt{C}_i$ is a normalizing constant to ensure $\sum_{k=1}^{K} \widetilde{\pi}_i[k] = 1$. Consequently, by taking $\mathtt{d}(\cdot, \cdot)$ as the Kullback–Leibler (KL) divergence $\mathtt{d}_{\mathrm{KL}}(\cdot, \cdot)$, the second loss term in (1) is defined as below:

$$\ell_{\mathrm{KL}}(\widetilde{p}_i; \widehat{\mathbf{y}}_i^{\mathrm{b}}, \widetilde{p}_i^{\mathrm{old}}) = \mathtt{d}_{\mathrm{KL}}(\widetilde{\pi}_i, \widetilde{p}_i) = \sum_{k=1}^{K} \widetilde{\pi}_i[k] \cdot \log\left(\frac{\widetilde{\pi}_i[k]}{\widetilde{p}_i[k]}\right) \tag{C5}$$

The associated batch-wise loss for batch $\mathcal{B}$ is:

$$\mathcal{L}_{\mathrm{KL}} = \frac{1}{|\mathcal{B}|} \sum_{i \in \mathcal{B}} \ell_{\mathrm{KL}}(\widetilde{p}_i; \widehat{\mathbf{y}}_i^{\mathrm{b}}, \widetilde{p}_i^{\mathrm{old}}). \tag{C6}$$

**Overall Training Objective.** Besides the MSE-based losses in Section 4.2, considering that feature scale plays a critical role in regression tasks (Chen et al., 2021), we optionally incorporate the following batch-wise feature norm regularization

---

**Algorithm 1** Step 1: Target Data Histogram Information Set Construction

---

**Input:** Source model $h_{\mathrm{S}} = f_{\mathrm{reg,S}} \circ g_{\mathrm{S}}$, target dataset $\mathcal{D}_{\mathrm{T}}$, dropout ratio $\rho$, number of sampling repetitions $M$

**Output:** Histogram bin number $K$, bin values $\{\widetilde{y}_1, \ldots, \widetilde{y}_K\}$, histogram information set $\mathcal{H} = \{(\mathrm{x}_i^{\mathrm{T}}, \widetilde{p}_i^{\mathrm{old}}, \widehat{\mathbf{y}}_i^{\mathrm{b}})\}_{i=1}^{N_{\mathrm{T}}}$

1: Initialize $\mathcal{H} \leftarrow \varnothing$
2: **for** each $\mathrm{x}_i^{\mathrm{T}} \in \mathcal{D}_{\mathrm{T}}$ **do**
3:     **for** $m = 1$ to $M$ **do**
4:         $\widehat{y}_{i,m} \leftarrow f_{\mathrm{reg,S}}(\mathtt{DROPOUT}(g_{\mathrm{S}}(\mathrm{x}_i^{\mathrm{T}}), \rho))$
5:     **end for**
6:     Collect uncertainty-aware predictions $\widehat{\mathbf{y}}_i = \{\widehat{y}_{i,1}, \ldots, \widehat{y}_{i,M}\}$
7: **end for**
8: Compute the number of histogram bins $K$ using Eq. (C1) based on $\{\widehat{\mathbf{y}}_i\}_{i=1}^{N_{\mathrm{T}}}$
9: Compute the representative bin values $\{\widetilde{y}_1, \ldots, \widetilde{y}_K\}$ using Eq. (C2)
10: **for** each $\mathrm{x}_i^{\mathrm{T}} \in \mathcal{D}_{\mathrm{T}}$ **do**
11:     Compute the empirical discrete distribution $\widetilde{p}_i^{\mathrm{old}}$ over $K$ bins as: $\widetilde{p}_i^{\mathrm{old}}[k] \triangleq \frac{1}{M} \sum_{m=1}^{M} \mathbb{I}(\widehat{y}_{i,m} \in \mathrm{Bin}_k)$ for $k \in [K]$
12:     Construct partial label (bin) set $\widehat{\mathbf{y}}_i^{\mathrm{b}}$ by mapping $\widehat{\mathbf{y}}_i$ to the associated bin indices
13:     Insert histogram information tuple $(\mathrm{x}_i^{\mathrm{T}}, \widetilde{p}_i^{\mathrm{old}}, \widehat{\mathbf{y}}_i^{\mathrm{b}})$ into $\mathcal{H}$
14: **end for**

---

term to encourage alignment between the feature norms of the target and source models, thereby promoting stable training dynamics and further guiding the learning of the regression model:

$$\mathcal{L}_{\mathrm{FN}} \triangleq \frac{1}{|\mathcal{B}|} \sum_{i \in \mathcal{B}} \big| \|g(\mathrm{x}_i^{\mathrm{T}})\| - \|g_{\mathrm{S}}(\mathrm{x}_i^{\mathrm{T}})\| \big|, \tag{C7}$$

where $g_{\mathrm{S}}(\cdot)$ denotes the fixed feature extractor obtained from the pre-trained source model, and $g(\cdot)$ represents the feature extractor being trained on the target domain.

Combining the components discussed above, the overall loss takes the following form, depending on whether $\mathcal{L}_{\mathrm{MSE}}$ in (5) or $\mathcal{L}_{\mathrm{RMSE}}$ in (7) is used:

$$\mathcal{L}_{\mathrm{MERCI}} = \underbrace{\lambda_{\mathrm{PL}}\mathcal{L}_{\mathrm{PL}} + \lambda_{\mathrm{prior}}\mathcal{L}_{\mathrm{KL}}}_{\mathcal{L}_{\mathrm{hist}}} + \underbrace{\lambda_{\mathrm{MSE}}\mathcal{L}_{\mathrm{MSE}} (+\lambda_{\mathrm{FN}}\mathcal{L}_{\mathrm{FN}})}_{\mathcal{L}_{\mathrm{reg}}}; \tag{C8a}$$

$$\mathcal{L}_{\mathrm{MERCI-R}} = \underbrace{\lambda_{\mathrm{PL}}\mathcal{L}_{\mathrm{PL}} + \lambda_{\mathrm{prior}}\mathcal{L}_{\mathrm{KL}}}_{\mathcal{L}_{\mathrm{hist}}} + \underbrace{\lambda_{\mathrm{RMSE}}\mathcal{L}_{\mathrm{RMSE}} (+\lambda_{\mathrm{FN}}\mathcal{L}_{\mathrm{FN}})}_{\mathcal{L}_{\mathrm{reg-r}}}, \tag{C8b}$$

where $\mathcal{L}_{\mathrm{PL}}$, $\mathcal{L}_{\mathrm{KL}}$, and $\mathcal{L}_{\mathrm{FN}}$ are given in (C4), (C6), and (C7), respectively, and $\lambda_{\mathrm{PL}}$, $\lambda_{\mathrm{prior}}$, $\lambda_{\mathrm{MSE}}/\lambda_{\mathrm{RMSE}}$, and $\lambda_{\mathrm{FN}}$ are weighting coefficients for the four components. To reduce the burden of manual hyperparameter tuning, these coefficients are treated as learnable parameters during training.

### C.2. Training Algorithm

In this subsection, we summarize the complete training procedure in Algorithms 1-2.

## D. Additional Experimental Details

### D.1. Details of Datasets

**UTKFace** (Zhang et al., 2017) is a human face dataset used for age estimation, consisting of 24,106 images with age labels ranging from 1 to 106. Each image is additionally annotated with gender and race information. We take gender as the domain attribute and conduct two domain adaptation tasks: Female → Male and Male → Female.

**Biwi-Kinect** (Fanelli et al., 2013) is a 3D face dataset designed for head pose estimation, comprising 15,678 images. Following the setup in previous work (Adachi et al., 2025), we use gender as the domain attribute and train three separate models to predict pitch, yaw, and roll angles, respectively. The label ranges are $[-70°, 80°]$ for yaw, $[-70°, 70°]$ for roll, and $[-90°, 60°]$ for pitch.

---

**Algorithm 2** Step 2: Joint Adaptation Training of $g$, $f_{\text{reg}}$, and $f_{\text{hist}}$

---

**Input:** Target data $\mathcal{D}_{\text{T}}$, histogram information set $\mathcal{H}$, source model $h_{\text{S}} = f_{\text{S}} \circ g_{\text{S}}$, number of epochs $\mathcal{T}$, number of warm-up epochs $\mathcal{T}_{\text{warm}}$, hyper-parameter $\alpha \in (0, 1)$

**Output:** Adapted target model $h_{\text{T}} = f_{\text{reg,T}} \circ g_{\text{T}}$

 1: Initialize target model $h_{\text{T}}$ with source feature extractor and regressor: $f_{\text{reg,T}} \leftarrow f_{\text{reg,S}}$, $g_{\text{T}} \leftarrow g_{\text{S}}$
 2: Initialize histogram head $f_{\text{hist}}$ with class number $K$
 3: Initialize $\lambda_{\text{PL}}$, $\lambda_{\text{prior}}$, $\lambda_{\text{MSE}}$ ($\lambda_{\text{RMSE}}$), and $\lambda_{\text{FN}}$
 4: **for** $t = 1$ to $\mathcal{T}$ **do**
 5:    **for** each mini-batch $\text{x}_i^{\text{T}} \in \mathcal{B}$ from $\mathcal{D}_{\text{T}}$ **do**
 6:       Obtain histogram prediction $\widetilde{p}_i \leftarrow f_{\text{hist}} \circ g_{\text{T}}(\text{x}_i^{\text{T}})$
 7:       `// Histogram loss`
 8:       Compute batch-wise partial label loss $\mathcal{L}_{\text{PL}}$ by Eq. (C3)–(C4)
 9:       Compute batch-wise KL divergence with the unimodal prior $\mathcal{L}_{\text{KL}}$ by Eq. (C5)–(C6)
10:       $\mathcal{L}_{\text{hist}} \leftarrow \lambda_{\text{PL}}\mathcal{L}_{\text{PL}} + \lambda_{\text{prior}}\mathcal{L}_{\text{KL}}$
11:       **if** $t \leq \mathcal{T}_{\text{warm}}$ **then**
12:          Update $f_{\text{hist}}$, $\lambda_{\text{PL}}$, and $\lambda_{\text{prior}}$ by minimizing $\mathcal{L}_{\text{hist}}$
13:       **else**
14:          `// Regression loss`
15:          Compute pseudo-label $\bar{y}_i$ based on $\widetilde{p}_i^{\text{old}}$ using Eq. (4)
16:          Compute batch-wise MSE $\mathcal{L}_{\text{MSE}}$ by Eq. (5), or RMSE $\mathcal{L}_{\text{RMSE}} \leftarrow \sqrt{\mathcal{L}_{\text{MSE}}}$
17:          Compute batch-wise feature norm loss $\mathcal{L}_{\text{FN}}$ by Eq. (C7)
18:          Compute regression loss:

$$\mathcal{L}_{\text{reg}} \leftarrow \lambda_{\text{MSE}}\mathcal{L}_{\text{MSE}} \left(+\lambda_{\text{FN}}\mathcal{L}_{\text{FN}}\right), \quad \mathcal{L}_{\text{reg}-\text{r}} \leftarrow \lambda_{\text{RMSE}}\mathcal{L}_{\text{RMSE}} \left(+\lambda_{\text{FN}}\mathcal{L}_{\text{FN}}\right)$$

19:          `// Overall batch-wise loss for MERCI (MERCI-R)`
20:          Compute $\mathcal{L}_{\text{MERCI}}$ ($\mathcal{L}_{\text{MERCI}-\text{R}}$) by Eq. (C8)
21:          Update $g_{\text{T}}$, $f_{\text{reg,T}}$, $f_{\text{hist}}$, $\lambda_{\text{PL}}$, $\lambda_{\text{prior}}$, $\lambda_{\text{MSE}}$ ($\lambda_{\text{RMSE}}$), and $\lambda_{\text{FN}}$ by minimizing $\mathcal{L}_{\text{MERCI}}$ ($\mathcal{L}_{\text{MERCI}-\text{R}}$)
22:       **end if**
23:    **end for**
24:    Update $\widetilde{p}_i^{\text{old}}$ in $\mathcal{H}$: $\widetilde{p}_i^{\text{old}} \leftarrow \texttt{NORMALIZE}\{\alpha\widetilde{p}_i^{\text{old}} + (1 - \alpha)\widetilde{p}_i\}$
25: **end for**

---

**California Housing** (Pace & Barry, 1997) is a tabular dataset used for house price prediction. Following prior work (He et al., 2024; Adachi et al., 2025), we define the source and target domains based on geographic regions (non-coastal vs. coastal). We conduct two domain adaptation tasks: Near Bay → Far Bay and Far Bay → Near Bay. The model architecture on California Housing consists of a five-layer MLP with one BatchNorm layer and ReLU activation functions.

**Digits** dataset includes two widely-used digit recognition datasets, **SVHN** (Netzer et al., 2011) and **MNIST** (LeCun et al., 1998). Although originally developed for classification, we follow the setup in Adachi et al. (2025) and train a regression model to directly predict the scalar value corresponding to each digit image.

The training and validation split ratio is set following Adachi et al. (2025).

### D.2. Details of SFDAR Problem

Distribution shift, a core problem in transfer learning and domain adaptation, has been widely studied under distinct theoretical frameworks and application scenarios (Ben-David et al., 2006; Fang et al., 2025; 2026b;a), with strong connections to diverse research topics including representation learning (Chen & Marchand, 2023), learning with label noise (Guo et al., 2023; 2024), data protection (Li et al., 2025; 2026) and fairness (Shui et al., 2022). In this section, we detail the types of distribution shift relevant to the SFDA setting.

**Distribution Shift Types.** In SFDA settings, the problem is typically formulated under joint distribution shift, where $P_{\text{xy}}^{\text{S}} \neq P_{\text{xy}}^{\text{T}}$. This assumption encompasses covariate shift (Ben-David et al., 2006; Mansour et al., 2009; Ben-David et al., 2010), label shift (Lipton et al., 2018; Garg et al., 2020; Guo et al., 2026), and posterior (or concept) shift (Cai & Wei, 2021;

Zhu et al., 2024; Maity et al., 2024), aligning with classical domain adaptation theory (Kouw & Loog, 2019) and joint alignment approaches (Long et al., 2013).

Heuristically guided by the decomposition $P_{xy} = P_x P_{y|x}$, current SFDA research often focuses on learning more robust representations or designing stronger supervisory signals for target adaptation (Liang et al., 2020), implicitly relying on two assumptions: (1) Similarity of feature extractors: the source-trained feature extractor produces semantically meaningful representations for target inputs (Yang et al., 2022b; Liang et al., 2020); (2) Proximity of conditional distributions: $P_{y|x}^S$ and $P_{y|x}^T$ are assumed to be different but not too far apart, which facilitates the use of output uncertainty for sample selection (Zhang et al., 2022b; Xu et al., 2025a).

Our method is developed under the above assumptions, with experiments conducted under a mixture of relaxed covariate (input) shift and label shift. To illustrate the range and distribution of the label space, Figures D.1 and D.2 present histograms of the ground-truth regression values across four datasets. These results demonstrate the diversity of label space distributions in the selected datasets, underscoring both the variety of experimental settings and the generalizability of the proposed method.

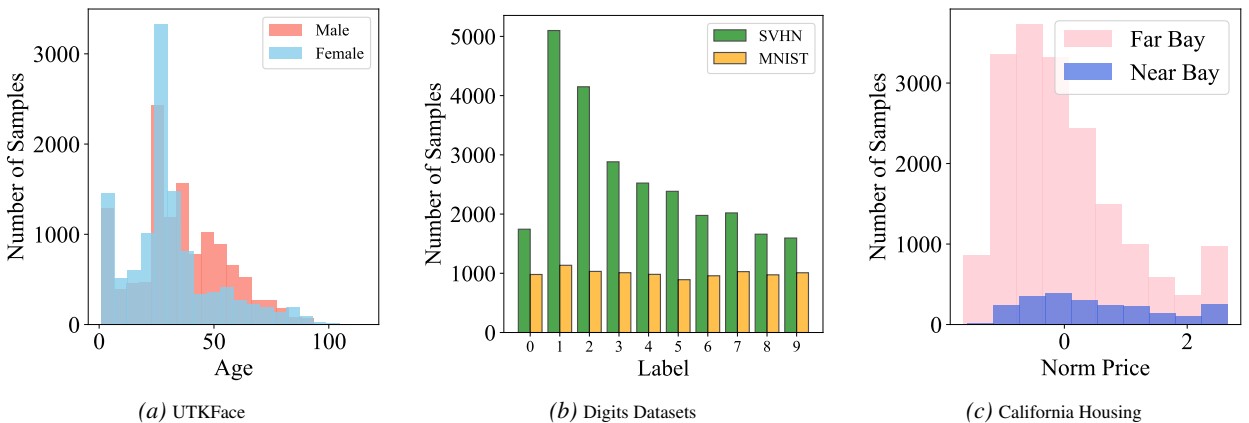

*(a)* UTKFace      *(b)* Digits Datasets      *(c)* California Housing

*Figure D.1. Histograms of Regression Labels for Different Domains Across Three Datasets*

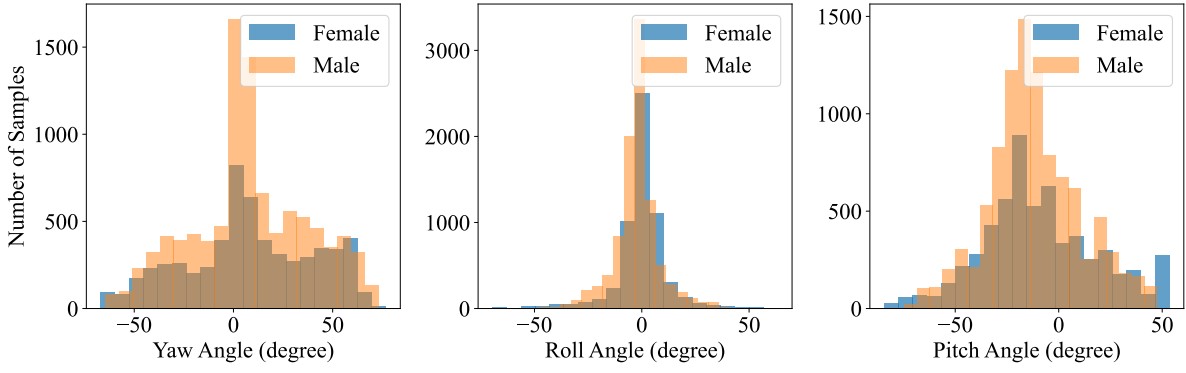

*Figure D.2. Histograms of Regression Labels for Different Domains on Biwi-Kinect Dataset*

**Importance and potential application scenarios of SFDAR.** Here, we aim to highlight the importance and practical relevance of the SFDAR problem. In fact, regression is a fundamental problem in machine learning and is closely related to numerous real-world applications, including object localization, image registration, and human pose estimation (Lathuilière et al., 2019).

While SFDA has been widely studied in classification and segmentation tasks (Guo et al., 2023; 2024; Yi et al., 2023), extending this paradigm to regression is both natural and necessary. For example, (1) in healthcare, tasks like predicting patient risk scores or tumor progression from imaging data often involve continuous outcomes. In a new hospital setting, source data may be inaccessible due to privacy concerns, and acquiring target labels requires expert annotations or follow-up

exams, making SFDAR particularly relevant. (2)In industrial applications, estimating the Remaining Useful Life (RUL) of machinery is a regression task where source data is often unavailable due to commercial confidentiality, and collecting target labels is costly or impractical, as it requires observing actual failure events.

The limited prior work underscores the methodological difficulty of this setting, which makes it both challenging and valuable to investigate. By formally defining the SFDAR setting and proposing concrete solutions, we aim to encourage broader interest and further exploration in this important yet underexplored domain.

### D.3. Evaluation Metrics

In this subsection, we introduce the detailed calculation of four most widely utilized regression evaluation metrics as follows:

MAE measures the average magnitude of absolute errors between predictions $y_{i,\text{pred}}$ and ground truth $y_{i,\text{true}}$:

$$\text{MAE} = \frac{1}{n} \sum_{i=1}^{n} |y_{i,\text{true}} - y_{i,\text{pred}}|$$

It provides an intuitive and direct estimate of the typical prediction error.

Compared to MAE, RMSE penalizes larger errors more heavily by squaring the residuals before averaging and more sensitive to outliers:

$$\text{RMSE} = \sqrt{\frac{1}{n} \sum_{i=1}^{n} (y_{i,\text{true}} - y_{i,\text{pred}})^2}.$$

The coefficient of determination, $R^2$, evaluates the proportion of variance in the target variable that is explained by the model's predictions:

$$R^2 = 1 - \frac{\sum_{i=1}^{n} (y_{i,\text{true}} - y_{i,\text{pred}})^2}{\sum_{i=1}^{n} (y_{i,\text{true}} - \bar{y}_{\text{true}})^2}$$

where $\bar{y}_{\text{true}}$ is the mean of the ground-truth values. A larger value means better performance and 1 indicates a perfect fit.

Finally, the Pearson correlation coefficient, R, measures the linear correlation between the predicted and true values:

$$R = \frac{\sum_{i=1}^{n} (y_{i,\text{true}} - \bar{y}_{\text{true}})(y_{i,\text{pred}} - \bar{y}_{\text{pred}})}{\sqrt{\sum_{i=1}^{n} (y_{i,\text{true}} - \bar{y}_{\text{true}})^2} \sqrt{\sum_{i=1}^{n} (y_{i,\text{pred}} - \bar{y}_{\text{pred}})^2}}$$

Values of $R$ range from -1 (perfect negative correlation) to 1 (perfect positive correlation), with 0 indicating no linear relationship.

MAE and RMSE are error-based metrics, while $R^2$ and R are correlation-based metrics. Together, these metrics provide a comprehensive assessment of both prediction accuracy and model reliability.

### D.4. Implementation Details

We use ResNet-26 (He et al., 2016) as the backbone for the Digits dataset, ResNet-50 for UTKFace and Biwi-Kinect, and a Multi-Layer Perceptron (MLP) for California Housing. A linear layer is used as the regression head, while MERCI adopts a bottleneck classification neural network with a 512-dimensional hidden layer for histogram learning (Liang et al., 2020). Following prior protocols, we train the source model for 100 epochs in both data-free and source-free settings. During adaptation, except for SSA, we perform full-parameter training for 30 epochs with a batch size of 64 using Adam. A smaller learning rate (0.0005) is applied to the feature extractor, and a larger one (0.005) to the regressor and classifier. In our method, loss coefficients are treated as learnable parameters and optimized using Adam with a learning rate of 0.005. All experiments are conducted using three random seeds on a single NVIDIA A100 GPU.

### D.5. Details of Baseline Methods

We introduce the implementation details of different baselines as follows:

**DANN** (Ganin et al., 2016) is one of the most classical domain adaptation method, designed for classification task but can be directly transferred into regression. By introducing a domain classifier and the adversarial training strategy, it encourage

domain-invariant feature learning, thereby aligning source and target distributions in the latent space. It jointly optimizes a label predictor and a domain discriminator through a gradient reversal layer.

**RSD** (Chen et al., 2021) and **DARE-GRAM** (Nejjar et al., 2023) are two unsupervised domain adaptation methods for regression that rely on representation space alignment and require simultaneous access to both source and target data.

**SSA** (Adachi et al., 2025) is a recently proposed test-time adaptation method based on feature alignment. Although SSA does not require access to raw source data during adaptation, it still relies on source data statistics (e.g., feature covariance matrices), which deviates from our source-free setting. We refer to this setting as a data-free domain adaptation task. Comparing the results in our setting with those reported in SSA (Adachi et al., 2025), we observe that SSA performs substantially better under covariate-shift scenarios and in shift-aware application settings. However, when facing mixed distribution shifts and changes in prediction scale, its performance tends to become less stable.

**TASFAR** (He et al., 2024) is another recently proposed data-free domain adaptation method designed for regression. It estimates sample-wise uncertainty through random sampling and refines regression labels accordingly. However, the method relies on the assumption of a specific distribution (e.g., Gaussian), and requires training an uncertainty predictor on source data. While TASFAR does not access raw source samples during adaptation, it still depends on source-trained components, making it incompatible with the strict source-free setting. One of its main limitations is the TASFAR's strong dependence on the quality of pseudo-labels, which may lead to performance instability, especially when the initial label estimation is inaccurate.

**BN-adapt** (Benz et al., 2021) is a widely used and effective source-free or test-time adaptation method. It is lightweight and well-suited for domain-adaptive regression tasks. It updates only the Batch Normalization layers' running statistics and does not require any backpropagation. Our experimental results confirm its strong performance, while also revealing that BN-adapt can be sensitive to the scale of the target response space.

**CRAFT** (Biswas et al., 2025) is a recently proposed semi-supervised method, whose unsupervised component (denoted as **CRAFT-U**) is directly applicable to the SFDAR setting. This method relies on an instance-level Gaussian assumption for the conditional target distribution and approximates the global label space using a Gaussian Mixture Model (GMM), resulting in a contrastive loss objective. We evaluated both full-parameter adaptation and BN-layer updating, and report the latter as it yielded superior performance.

**BRR** (Zhan et al., 2025) is the first formally proposed method that strictly addresses SFDAR problems. It tackles the SFDAR problem via self-training-based alignment at both the feature and label levels, without requiring access to source data. For feature distribution alignment, BRR discretizes the continuous target space into bins; however, these bins are primarily used for feature smoothing or confidence score assignment, rather than explicitly modeling and correcting target label distribution. In addition, the original experiments of BRR are conducted on tabular regression tasks. As no official implementation is publicly available, we reimplement BRR and evaluate multiple adaptation configurations, including full-parameter adaptation, BN-layer updating, and regression head updating. We report results obtained under BN-layer adaptation, which consistently yields superior performance among these settings.

To further validate the effectiveness of our approach, we propose two additional data augmentation-based SFDAR methods, VM and AugSelfTr. Inspired by prior works (Adachi et al., 2025; Zhang et al., 2022a), both methods aim to minimize the variance of model outputs under data augmentation. Specifically, we apply pre-generated augmentations (AugMix (Hendrycks et al., 2020)) in **AugSelfTr**, and on-the-fly augmentations in **VM** (Chen et al., 2020).

**Implementation Details of Baseline Models.** For UDA and DF methods, we conduct experiments using their officially released code. For UDA methods, we perform adaptation training for 50 epochs. We briefly tune the hyperparameters for each method and report the best-performing results. Specifically, we adjust `rsd_coef` and `bmp_coef` for RSD, and `threshold`, `scale_coef` and `angle_coef` for DARE-GRAM. For the test-time adaptation method SSA, we follow the original protocol by forwarding the entire dataset once and updating only the BatchNorm layers, which yields optimal performance. For the semi-supervised method CRAFT (Biswas et al., 2025), we adopt its unsupervised component (CRAFT-U) and apply a small scaling coefficient ($10^{-4}$) to the loss term for computational stability. We tuned the hyperparameters `n_bins` and `n_comps` and report the best performance. For BRR, we evaluate different parameter-updating and feature-updating strategies, and report the best performance achieved when both the Batch Normalization layers and stored feature representations are updated. For TASFAR, VM, and AugSelfTr, we train all parameters for 30 epochs to ensure a fair comparison.

# E. Additional Experimental Results

## E.1. Additional Regression Results

Due to space limitations in the main text, we present the results of four regression metrics on the SVHN→MNIST and MNIST→SVHN adaptation tasks from the Digits dataset in Figure E.2. Additionally, the **RMSE** and $R$ scores on the Biwi-Kinect dataset are reported in Tables E.3 and E.4.

We further evaluate MERCI on the larger-scale MPI3D dataset (Gondal et al., 2019) with over one million images across three domains: Real (RL), Realistic (RC), and Toy (T). We implement the experiments on a subset of 100,000 images. Following RSD and DARE-GRAM, we consider two regression tasks (rotation about a vertical and horizontal axis) and report the sum of MAE. On this large-scale dataset, the performance gap between UDA and SFDA methods is more pronounced, as richer source statistics benefit UDA alignment. Nevertheless, **MERCI-R achieves the best performance among all SFDA methods**. BRR's fixed discretization of continuous labels introduces quantization errors that accumulate at scale, leading to degraded performance even relative to BN-adapt. Overall, the results in Tables E.5 and E.6 show that MERCI generalizes well to larger datasets, with a moderate and acceptable increase in computational cost.

Table E.2. Results on **Digits** datasets. Best non-Oracle SF results are **bolded**, best non-Oracle results are underlined.

| Method | Type | MNIST → SVHN | | | | SVHN → MNIST | | | |
|---|---|---|---|---|---|---|---|---|---|
| | | MAE ↓ | RMSE ↓ | $R^2$ ↑ | R ↑ | MAE ↓ | RMSE ↓ | $R^2$ ↑ | R ↑ |
| DANN (Ganin et al., 2016) | UDA | 2.2290 | 2.8487 | -0.1330 | 0.1974 | 1.0980 | 1.7586 | 0.6308 | 0.8025 |
| RSD (Chen et al., 2021) | UDA | 2.2054 | 2.8162 | -0.1067 | 0.2329 | 1.0528 | 1.7917 | 0.6157 | 0.7950 |
| DARE-GRAM (Nejjar et al., 2023) | UDA | 2.1480 | 2.7968 | -0.0914 | 0.2680 | 1.1707 | 1.8072 | 0.6105 | 0.7853 |
| SSA (Adachi et al., 2025) | DF | 2.6255 | 3.5262 | -0.7360 | 0.1364 | 1.3734 | 2.1633 | 0.4395 | 0.6911 |
| TASFAR (He et al., 2024) | DF | 2.8918 | 3.7616 | -0.9969 | 0.0845 | 1.6768 | 2.2270 | 0.4024 | 0.6426 |
| Source | SF | 2.9124 | 3.7935 | -1.0305 | 0.1043 | 1.7155 | 2.3422 | 0.3419 | 0.5951 |
| BN-adapt (Benz et al., 2021) | SF | 2.5836 | 3.4625 | -0.6732 | 0.1335 | 1.4787 | 2.2426 | 0.3955 | 0.6465 |
| CRAFT-U (Biswas et al., 2025) | SF | 2.4954 | 3.3037 | -0.5238 | 0.1480 | 1.5025 | 2.2578 | 0.3872 | 0.6397 |
| BRR (Zhan et al., 2025) | SF | 2.4581 | 3.3351 | -0.5519 | 0.1783 | 1.5101 | 2.2585 | 0.3870 | 0.6375 |
| VM | SF | 2.8228 | 3.7007 | -0.9343 | 0.1213 | 1.6513 | 2.2604 | 0.3849 | 0.6246 |
| AugSelfTr | SF | 2.8423 | 3.7073 | -0.9436 | 0.1178 | 1.5995 | 2.1078 | 0.4632 | 0.6897 |
| MERCI | SF | **2.4378** | **3.2164** | **-0.4590** | 0.1623 | **1.4770** | **1.9750** | **0.5290** | **0.7665** |
| MERCI w. FN | SF | 2.4593 | 3.2423 | -0.4830 | **0.1814** | 1.4971 | 1.9982 | 0.5183 | 0.7481 |
| MERCI-R | SF | 2.4808 | 3.2591 | -0.4952 | 0.1411 | 1.5090 | 1.9852 | 0.5172 | 0.7376 |
| MERCI-R w. FN | SF | 2.5207 | 3.3101 | -0.5447 | 0.1698 | 1.5159 | 2.0138 | 0.5100 | 0.7534 |
| Oracle | - | 0.3520 | 1.0320 | 0.8514 | 0.9235 | 0.0631 | 0.3251 | 0.9874 | 0.9937 |

Table E.3. **RMSE** on **Biwi-Kinect** dataset (Fanelli et al., 2013). Best non-Oracle SF results are **bolded**, best non-Oracle results are underlined.

| Method | Type | Female → Male | | | | Male → Female | | | |
|---|---|---|---|---|---|---|---|---|---|
| | | Pitch | Roll | Yaw | Mean | Pitch | Roll | Yaw | Mean |
| DANN (Ganin et al., 2016) | UDA | 6.2469 | 6.7258 | 4.4961 | 5.8229 | 10.0889 | 7.7898 | 8.6000 | 8.8262 |
| RSD (Chen et al., 2021) | UDA | 6.2309 | 6.6824 | 4.4833 | 5.7989 | 10.4134 | 7.6194 | 8.2568 | 8.7632 |
| DARE-GRAM (Nejjar et al., 2023) | UDA | 6.3410 | 6.8664 | 4.5956 | 5.9343 | 10.0607 | 7.7526 | 7.6536 | 8.4890 |
| SSA (Adachi et al., 2025) | DF | 6.7116 | 7.0266 | 4.6490 | 6.1291 | 10.1474 | 8.0121 | 8.5636 | 8.9077 |
| TASFAR (He et al., 2024) | DF | 6.8092 | 7.3885 | 4.7654 | 6.3210 | 12.5730 | 8.5134 | 8.8511 | 9.9792 |
| Source | SF | 6.7130 | 7.1649 | 4.6409 | 6.1729 | 12.2213 | 8.4473 | 8.7855 | 9.8180 |
| BN-adapt (Benz et al., 2021) | SF | 6.7960 | 7.1436 | 4.6135 | 6.1843 | 11.5047 | 8.1348 | 8.3536 | 9.3310 |
| CRAFT-U (Biswas et al., 2025) | SF | 6.8183 | 7.1757 | 4.5929 | 6.1956 | 11.3611 | 8.1189 | 8.3232 | 9.2677 |
| BRR (Zhan et al., 2025) | SF | 6.8261 | 7.0627 | 4.6034 | 6.1640 | 11.3354 | 8.2007 | 8.3732 | 9.3031 |
| VM | SF | 6.5037 | 7.0540 | 4.4677 | 6.0085 | 11.9837 | 8.1452 | 8.5477 | 9.5589 |
| AugSelfTr | SF | 6.6517 | 7.0615 | 4.4441 | 6.0524 | 11.4350 | 7.9532 | 7.8298 | 9.0726 |
| MERCI | SF | 6.2487 | 7.0180 | 4.4659 | 5.9108 | 10.9115 | 7.9664 | 7.0137 | 8.6306 |
| MERCI w. FN | SF | 6.2111 | 6.9568 | 4.4298 | 5.8659 | **10.6790** | 7.8997 | 6.5044 | 8.3610 |
| MERCI-R | SF | **6.1018** | **6.9321** | **4.3859** | **5.8066** | 11.1503 | 7.9469 | 7.1924 | 8.7632 |
| MERCI-R w. FN | SF | 6.1992 | 6.9554 | 4.3939 | 5.8495 | 10.7193 | **7.8149** | **6.2131** | **8.2491** |
| Oracle | - | 1.1479 | 1.4720 | 1.1462 | 1.2554 | 1.6862 | 1.9784 | 1.5135 | 1.7260 |

*Table E.4.* **R** on **Biwi-Kinect** dataset (*Fanelli et al., 2013*). Best non-Oracle SF results are **bolded**, best non-Oracle results are underlined.

| Method | Type | Female → Male | | | | Male → Female | | | |
|---|---|---|---|---|---|---|---|---|---|
| | | Pitch | Roll | Yaw | Mean | Pitch | Roll | Yaw | Mean |
| DANN (Ganin et al., 2016) | UDA | 0.9642 | 0.7939 | 0.9882 | 0.9154 | 0.9445 | 0.8236 | 0.9796 | 0.9159 |
| RSD (Chen et al., 2021) | UDA | 0.9641 | 0.7948 | 0.9884 | 0.9158 | 0.9368 | 0.8333 | 0.9803 | 0.9168 |
| DARE-GRAM (Nejjar et al., 2023) | UDA | 0.9624 | 0.7730 | 0.9879 | 0.9078 | 0.9420 | 0.8121 | 0.9782 | 0.9108 |
| SSA (Adachi et al., 2025) | DF | 0.9612 | 0.7908 | 0.9879 | 0.9133 | 0.9529 | 0.8201 | 0.9826 | 0.9185 |
| TASFAR (He et al., 2024) | DF | 0.9601 | 0.7390 | 0.9850 | 0.8947 | 0.9277 | 0.7857 | 0.9600 | 0.8911 |
| Source | SF | 0.9608 | 0.7658 | 0.9878 | 0.9048 | 0.9229 | 0.7906 | 0.9789 | 0.8975 |
| BN-adapt (Benz et al., 2021) | SF | 0.9615 | 0.7792 | 0.9876 | 0.9094 | 0.9304 | 0.8139 | 0.9809 | 0.9084 |
| CRAFT-U (Biswas et al., 2025) | SF | 0.9610 | 0.7802 | 0.9885 | 0.9099 | 0.9303 | 0.8132 | 0.9803 | 0.9079 |
| BRR (Zhan et al., 2025) | SF | 0.9619 | **0.7951** | 0.9885 | **0.9152** | **0.9331** | 0.8185 | 0.9806 | 0.9108 |
| VM | SF | 0.9634 | 0.7728 | 0.9884 | 0.9082 | 0.9240 | 0.8101 | 0.9791 | 0.9044 |
| AugSelfTr | SF | 0.9621 | 0.7651 | 0.9888 | 0.9053 | 0.9264 | 0.8178 | 0.9788 | 0.9077 |
| MERCI | SF | **0.9646** | 0.7639 | 0.9886 | 0.9057 | 0.9273 | 0.8181 | 0.9804 | 0.9086 |
| MERCI w. FN | SF | 0.9640 | 0.7723 | 0.9886 | 0.9083 | 0.9319 | 0.8255 | 0.9814 | 0.9129 |
| MERCI-R | SF | 0.9640 | 0.7695 | 0.9890 | 0.9075 | 0.9283 | 0.8189 | 0.9814 | 0.9095 |
| MERCI-R w. FN | SF | 0.9645 | 0.7675 | **0.9892** | 0.9071 | 0.9316 | **0.8288** | **0.9830** | **0.9144** |
| Oracle | - | 0.9988 | 0.9911 | 0.9993 | 0.9964 | 0.9982 | 0.9888 | 0.9989 | 0.9953 |

*Table E.5.* Results (**MAE**) on MPI3D, a **large-scale benchmark** with over one million images across three domains: *Real (RL), Realistic (RC), and Toy (T)*.

| Methods | Type | RL→RC | RL→T | RC→RL | RC→T | T→RL | T→RC | Avg |
|---|---|---|---|---|---|---|---|---|
| Resnet-18 | UDA | 0.17 | 0.44 | 0.19 | 0.45 | 0.51 | 0.50 | 0.377 |
| TCA | UDA | 0.17 | 0.42 | 0.19 | 0.42 | 0.50 | 0.50 | 0.373 |
| MCD | UDA | 0.13 | 0.40 | 0.15 | 0.45 | 0.52 | 0.50 | 0.358 |
| JDOT | UDA | 0.16 | 0.41 | 0.16 | 0.41 | 0.47 | 0.47 | 0.353 |
| AFN | UDA | 0.18 | 0.45 | 0.20 | 0.46 | 0.53 | 0.53 | 0.390 |
| DAN | UDA | 0.12 | 0.35 | 0.12 | 0.27 | 0.40 | 0.41 | 0.278 |
| DANN | UDA | 0.09 | 0.24 | 0.11 | 0.41 | 0.48 | 0.37 | 0.283 |
| RSD | UDA | 0.09 | 0.19 | **0.08** | 0.15 | 0.36 | 0.36 | 0.205 |
| DARE-GRAM | UDA | **0.09** | **0.15** | 0.10 | **0.14** | **0.24** | **0.24** | **0.160** |
| BN-adapt | SFDA | 0.19 | 0.44 | 0.19 | 0.43 | 0.46 | 0.44 | 0.359 |
| BRR | SFDA | 0.26 | 0.46 | 0.25 | 0.44 | 0.47 | 0.46 | 0.392 |
| MERCI-R | SFDA | **0.15** | **0.38** | **0.14** | **0.36** | **0.42** | **0.41** | **0.308** |

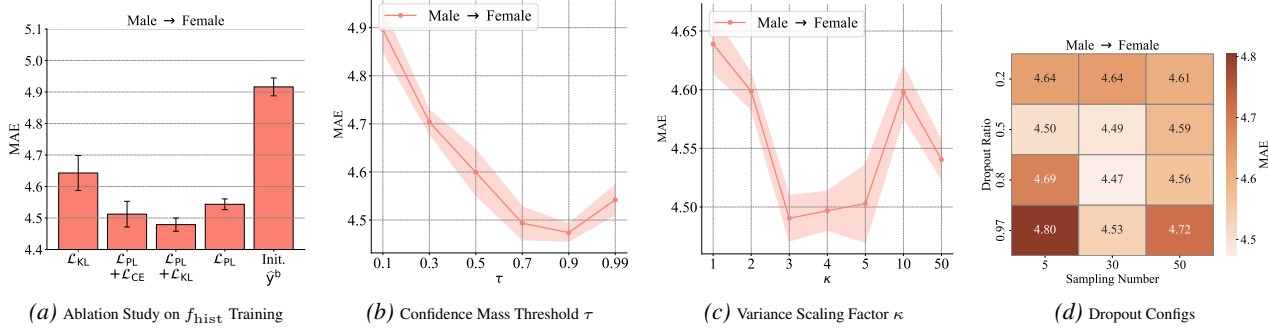

*(a)* Ablation Study on $f_{\text{hist}}$ Training    *(b)* Confidence Mass Threshold $\tau$    *(c)* Variance Scaling Factor $\kappa$    *(d)* Dropout Configs

*Figure E.3.* Ablation and hyperparameter sensitivity analysis on the Biwi-Kinect dataset (Male → Female, Yaw)

## E.2. Additional Ablation Results and Discussions

**Ablation Results.** Similar to the ablation study of the histogram head presented in the main paper, we conduct experiments

Table E.6. Efficiency comparison of MERCI and baseline methods on large-scale dataset MPI3D.

| Method | Init Time (s) | Train Time / Epoch (s) | Peak GPU Memory |
|---|---|---|---|
| BN-adapt | 0.00 | 1090.32 | 0.49 GB |
| BRR | 843.68 | 1170.82 | 1.77 GB |
| MERCI-R | 1780.21 | 1261.87 | 1.99 GB |

Table E.7. Ablation Study on **Moving Average Coefficient ($\alpha$) Sensitivity (UTKFace)**.

| Moving Average Coef. $\alpha$ | 0.01 | 0.1 | 0.3 | 0.5 | 0.7 | 0.9 | 0.99 |
|---|---|---|---|---|---|---|---|
| SOURCE (MAE), female→male | 6.5494 | 6.5494 | 6.5494 | 6.5494 | 6.5494 | 6.5494 | 6.5494 |
| MERCI-R (MAE), female→male | 6.1334 | 6.0969 | 5.9316 | 5.8263 | 5.8590 | 5.9473 | 6.0270 |
| SOURCE (MAE), male→female | 7.9390 | 7.9390 | 7.9390 | 7.9390 | 7.9390 | 7.9390 | 7.9390 |
| MERCI-R (MAE), male→female | 7.2304 | 7.1785 | 7.1577 | 7.0802 | 7.0740 | 7.0157 | 7.0780 |

on the Biwi-Kinect dataset under the Male → Female, as shown in Figure E.3a.

**Discussion about $\mathcal{L}_{\mathrm{FN}}$.** Inspired by prior work (Chen et al., 2021), we note that feature scale is important in regression tasks and may affect adaptation performance. In the Source-Free Domain Adaptive Regression setting, the source model already yields informative predictions on the target domain. To continue benefiting from the source model while ensuring stable and robust adaptation, we introduce the FN term to regularize the feature scale. In practice, we find that while the FN loss helps maintain a consistent feature scale, it can also restrict the extent of updates to the target model, potentially limiting performance. In cases where the target model can substantially outperform the source model after adaptation (e.g., UTKFace), the gain from FN is marginal. In contrast, for more challenging adaptation tasks (e.g., California Housing), the FN loss offers more noticeable improvements, as shown in Table 2 in the main paper. Additionally, our overall framework, MERCI, already improves adaptation robustness, which reduces the relative importance and observable benefit of the FN term in some scenarios.

### E.3. Addiional Hyperparameters Sensitivity Analysis

**$\tau$, $\kappa$ and Dropout Configuration.** Similar to the hyperparameter sensitivity analysis presented in the main paper, we conduct experiments on the Biwi-Kinect dataset under the Male → Female, as shown in Figure E.3b- E.3d.

**Moving Average Coefficient $\alpha$.** Following previous work (Qiu et al., 2021), we adopt a moving average strategy to update the empirical discrete distribution, $\widetilde{p}_i^{\mathrm{old}}$, in the Histogram Information Set, which stabilizes the adaptation process and preserves useful information from the source model predictions.

In our implementation, we fix the update ratio to 0.5. To further validate the effect of the moving average strategy on histogram head training and the resulting discrete distribution, we conduct additional ablation studies, as shown in Table E.7. When $\alpha$ is small, the histogram used to generate pseudo-labels for the regressor is primarily influenced by the current epoch's output. As $\alpha$ increases, the histogram incorporates more information from the original partial-label set and the prior. Within a broad range of values (0.3–0.9), the experimental results demonstrate stable performance, confirming the both the practicality and robustness of the moving average mechanism.

**Loss Coefficients.** In our experiments, the weighting coefficients are treated as learnable parameters. This design choice aligns the source-free domain adaptation setting, where tuning or selecting hyperparameters manually is often impractical in real-world applications.

To avoid the trivial solution such that $\lambda \to 0$, we follow the well-established uncertainty-based multi-task weighting formulation (Kendall et al., 2018) and take the following steps. First, each weight is parameterized as $\lambda = \exp(-\log \sigma^2)$, and the total loss contains the regularization term $\frac{1}{2} \sum_i \log \sigma_i^2$, which increases when $\lambda$ becomes too small, thereby preventing collapse. Second, we clamp $\log \sigma_i^2 \in [-10, 10]$ to keep all weights within a stable, non-zero range. Third, NaN/Inf values revert to uniform weighting for numerical safety. The learning weight is reset at the beginning of each epoch, and the training dynamics on the UTKFace and Biwi-Kinect datasets are illustrated in Figure E.4.

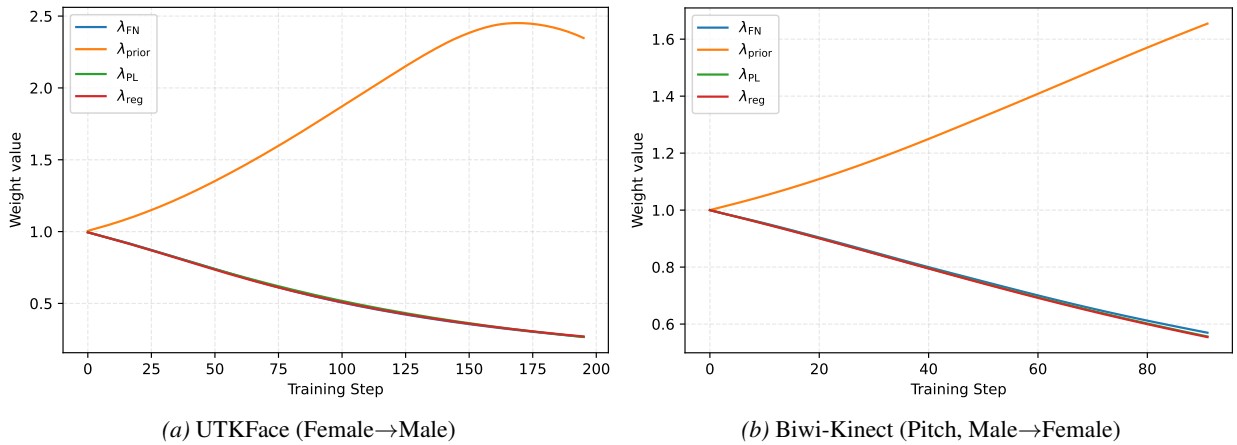

*(a)* UTKFace (Female→Male)  *(b)* Biwi-Kinect (Pitch, Male→Female)

*Figure E.4. Training trajectories of the learned loss weights on UTKFace and Biwi-Kinect. In both datasets, the weights evolve smoothly and remain within stable, non-zero ranges, showing that our uncertainty-based formulation and regularization prevent the trivial collapse $\lambda \to 0$.*

*Table E.8. Optimized $\boldsymbol{\lambda}$ Values for Each Dataset and Shift Setting.*

| Dataset | Source → Target | $\lambda_{\mathrm{reg}}$ | $\lambda_{\mathrm{PL}}$ | $\lambda_{\mathrm{prior}}$ | $\lambda_{\mathrm{FN}}$ |
|---|---|---|---|---|---|
| UTKface | female → male | 0.26 | 0.27 | 2.31 | 0.26 |
| UTKface | male → female | 0.30 | 0.30 | 1.95 | 0.32 |
| Biwi Kinect | female → male (pitch) | 0.37 | 0.36 | 1.15 | 0.38 |
| Biwi Kinect | female → male (yaw) | 0.36 | 0.36 | 1.74 | 0.36 |
| Biwi Kinect | female → male (roll) | 0.37 | 0.36 | 1.23 | 0.40 |
| Biwi Kinect | male → female (pitch) | 0.55 | 0.56 | 1.53 | 0.58 |
| Biwi Kinect | male → female (yaw) | 0.56 | 0.56 | 1.64 | 0.56 |
| Biwi Kinect | male → female (roll) | 0.56 | 0.56 | 1.78 | 0.75 |
| Digits | SVHN → MNIST | 1.09 | 0.35 | 2.87 | 2.99 |
| Digits | MNIST → SVHN | 2.26 | 0.08 | 1.31 | 1.45 |
| California House | Near Bay → Far Bay | 1.00 | 1.00 | 1.00 | 1.00 |
| California House | Far Bay → Near Bay | 1.00 | 1.00 | 1.00 | 1.00 |

To provide further guidance, we report the final optimized values in Table E.8. Notably, while most weights remain similar, the $\lambda_{\mathrm{prior}}$ values vary across datasets. To assess the sensitivity of performance to this variation, we conducted additional ablation studies on the UTKFace dataset. As shown in Table E.9, our method remains robust to different $\lambda_{\mathrm{prior}}$ values.

*Table E.9. Ablation Study (MAE) on $\boldsymbol{\lambda}_{\mathbf{prior}}$ Sensitivity (UTKFace).*

| Setting (($\lambda_{\mathrm{reg}}$-$\lambda_{\mathrm{PL}}$-$\lambda_{\mathrm{prior}}$-$\lambda_{\mathrm{FN}}$)) | UTKface (female→male) | UTKface (male→female) |
|---|---|---|
| 0.3-0.3-0.3-0.3 | 6.00 | 7.09 |
| 0.3-0.3-0.8-0.3 | 5.95 | 7.19 |
| 0.3-0.3-1.0-0.3 | 5.99 | 7.11 |
| 0.3-0.3-2.0-0.3 | 5.96 | 7.05 |
| 0.3-0.3-3.0-0.3 | 6.01 | 7.16 |
| 0.3-0.3-4.0-0.3 | 5.95 | 7.27 |
| 1.0-1.0-1.0-1.0 | 5.99 | 7.02 |
| Learnable | 5.86 | 7.18 |

*Table E.10. Ablation Study (**MAE**) on **Bin Number** (**UTKFace**).*

| Bin Number | 3 | 5 | 10 | 15 | 25 | 45 | 65 | 85 | 100 | Self-Adaptive |
|---|---|---|---|---|---|---|---|---|---|---|
| UTKFace (female → male) | 23.98 | 14.54 | 7.92 | 6.55 | 5.97 | 5.82 | 5.88 | 5.83 | 5.84 | 5.83 |
| Histogram Head GT (female → male) | 23.65 | 14.69 | 6.51 | 4.74 | 2.74 | 1.73 | 0.91 | 0.70 | 0.70 | 1.84 |
| UTKFace (male → female) | 17.09 | 12.74 | 8.66 | 7.94 | 7.05 | 7.06 | 7.08 | 7.33 | 7.00 | 7.08 |
| Histogram Head GT (male → female) | 18.09 | 13.16 | 6.82 | 4.37 | 2.78 | 1.10 | 0.87 | 0.78 | 0.69 | 0.87 |

*Table E.11. Comparison between pseudo-label **MAE** (derived from the learned density/histogram) and discretization-induced error (computed on ground-truth). The **discretization error is consistently much smaller**, suggesting that the information loss induced by discretization is not a dominant factor compared to model and domain shift errors.*

| Dataset | Domains | Final MAE ($\bar{y}$) | Discretized GT MAE |
|---|---|---|---|
| UTKFace | Female → Male | 5.77 | 1.64 |
| UTKFace | Male → Female | 6.94 | 1.03 |
| Biwi-Kinect (Pitch) | Female → Male | 4.53 | 1.98 |
| Biwi-Kinect (Yaw) | Female → Male | 3.39 | 1.76 |
| Biwi-Kinect (Roll) | Female → Male | 4.87 | 0.51 |
| Biwi-Kinect (Pitch) | Male → Female | 7.40 | 1.88 |
| Biwi-Kinect (Yaw) | Male → Female | 5.16 | 1.84 |
| Biwi-Kinect (Roll) | Male → Female | 5.07 | 1.07 |
| Digits | SVHN → MNIST | 1.51 | 0.37 |
| Digits | MNIST → SVHN | 2.49 | 0.27 |
| California Housing | Near Bay → Far Bay | 0.66 | 0.19 |
| California Housing | Far Bay → Near Bay | 0.47 | 0.11 |

**Bin Size.** In the SFDA setting, selecting an optimal bin size is also challenging due to the absence of labeled target data. To address this, we propose a self-adaptive discretization strategy that automatically estimates the number of bins, with results shown in Table 3 in the main paper.

To provide further insight, we conduct an ablation study on the number of bins using the UTKFace dataset (Table E.10). The results show that performance remains generally stable across a broad range of bin counts but degrades notably when the number of bins is extremely small (e.g., fewer than 10). In such cases, the histogram becomes too coarse to faithfully represent the label distribution, introducing substantial quantization error. This, in turn, degrades the quality of both the partial-label supervision and KL regularization, and may propagate inaccurate pseudo-labels to the regressor, ultimately leading to performance collapse. Importantly, our self-adaptive method effectively avoids this low bin number failure regime by selecting a data-driven bin count, achieving competitive performance without manual tuning.

**Controlled discretization error.** Discretization introduces a quantization error $|\hat{y}_m - \hat{y}_m^b| < \frac{\Delta}{2}$, where the bin width $\Delta = \frac{\text{range}(\mathcal{Y})}{K}$ is adaptively controlled (Appendix C.1). In practice, this error is small compared to the **overall adaptation error**. For example, on Biwi-Kinect, the quantization error (1.42) is substantially smaller than the adaptation MAE (4.3), indicating that discretization is not the primary performance bottleneck. The full statistics across all datasets are provided in Table E.11.

### E.4. Performance on Non-Overlapping Label Space

To further evaluate the generality of our method, we examine MERCI's capacity to handle non-overlapping source and target label spaces. Specifically, we evaluate on UTKFace with disjoint label spaces (Female: y < 35, Male: y > 35). As shown in Table E.12, MERCI-R consistently outperforms SFDA baselines (BN-Adapt and BRR) in both directions, as the histogram is built from target-domain stochastic predictions that enable uncertainty-aware extrapolation beyond the source label range or fixed discretization. These results confirm that the histogram-based design is particularly effective in non-overlapping settings.

*Table E.12*

| Method | Female→Male (MAE)↓ | Male→Female (MAE)↓ |
|---|---|---|
| DARE-GRAM (UDA) | 25.99 | 11.99 |
| BN-Adapt (SFDA) | 29.03 | 27.31 |
| BRR (SFDA) | 24.49 | 20.35 |
| **MERCI-R (Ours)** | **20.67** | **17.18** |

*Table E.13. Performance (**MAE**) under increasing domain shift severity on UTKFace, simulated by Gaussian noise. MERCI consistently outperforms BRR across all shift levels, and the improvement becomes more pronounced under larger shifts. Combining MERCI with BatchNorm (MERCI + BN-adapt) further improves performance, indicating that MERCI is complementary to feature-level adaptation.*

| Domain | Method | N/A | Light | Mild | Severe | Ext. Sev. |
|---|---|---|---|---|---|---|
| Female → Male | SOURCE | 6.549 | 12.635 | 15.667 | 19.067 | 22.082 |
| | MERCI-R | 5.826 | 9.081 | 11.837 | 15.554 | 21.396 |
| | BRR | 6.229 | 9.883 | 13.417 | 16.462 | 21.809 |
| | BN-adapt | 6.302 | 9.048 | 10.221 | 11.889 | 15.520 |
| | **MERCI-R + BN-adapt** | - | **8.615** | **9.444** | **10.600** | **13.976** |
| Male → Female | SOURCE | 7.939 | 11.374 | 15.261 | 21.927 | 23.686 |
| | MERCI-R | 7.080 | 9.995 | 12.137 | 14.775 | 20.302 |
| | BRR | 7.201 | 10.761 | 13.073 | 15.526 | 20.324 |
| | BN-adapt | 7.339 | 9.644 | 10.714 | 12.093 | 14.413 |
| | **MERCI-R + BN-adapt** | - | **8.761** | **9.605** | **10.771** | **13.227** |

*Table E.14. Performance of MERCI-R on Toy Data with **Shift Severity (MAE)**.*

| Shift Severity | 0.1 | 0.15 | 0.2 | 0.25 | 0.375 | 0.5 | 1 | 1.5 |
|---|---|---|---|---|---|---|---|---|
| MERCI-R (MAE) | 0.168 | 0.304 | 0.625 | 0.882 | 1.265 | 1.716 | 3.279 | 4.704 |
| Source (MAE) | 0.318 | 0.518 | 0.813 | 1.008 | 1.482 | 1.926 | 3.530 | 4.927 |
| Avg. Partial Label Set Size | 4.083 | 4.101 | 4.133 | 4.190 | 4.285 | 4.670 | 6.102 | 7.565 |

## E.5. Robustness of MERCI under Different Levels of Distribution Shift

To evaluate MERCI's performance under substantial distribution shifts, we consider two settings: (1) A real-world dataset with Gaussian noise, where larger mean values induce stronger shifts; (2) A 2D toy dataset with polynomial-based inputs and controllable linear shifts.

The corresponding results are reported in Table E.13 (UTKFace) and Table E.14 (Toy Dataset). For Table E.13, we introduce Gaussian noise into the target domain at varying severity levels using the public corruption library *imagenet-c*. This library controls noise via a severity parameter ranging from 0 to 5. A severity of "N/A" indicates no added noise, so only natural domain shift is present; severity 1 corresponds to a "light" setting, severity 2 to "mild", severity 3 to "severe", and severity 5 to "extremely severe". Each severity level maps to a predefined standard deviation of Gaussian noise (e.g., 0.08 for severity 1 and 0.38 for severity 5), allowing us to simulate progressively stronger domain shifts in a controlled manner. As for Table E.14, shift severity is quantified by the central distance between source and target inputs.

The proposed MERCI framework demonstrates overall robustness across different shift magnitudes and performs well under slight to moderate shifts. In extreme shift scenarios, MERCI's reliance on the source model's predictions for target data limits its performance. Specifically, it struggles to generate reliable estimates of $\hat{y}^b$ when the domain gap is too large. This reflects a realistic constraint: in the absence of source data, adaptation becomes fundamentally challenging when the target domain diverges significantly from the source. Furthermore, comparing *MERCI-R*, *BN-adapt*, and *MERCI-R+BN-adapt* in Table E.13 reveals that MERCI-R's performance can be further improved through complementary stabilization techniques such as BatchNorm adaptation. Nevertheless, MERCI provides uncertainty estimates for pseudo-labels—for instance, the average length of partial label sets (as shown in Table E.14), which can serve as a proxy indicator of shift severity.

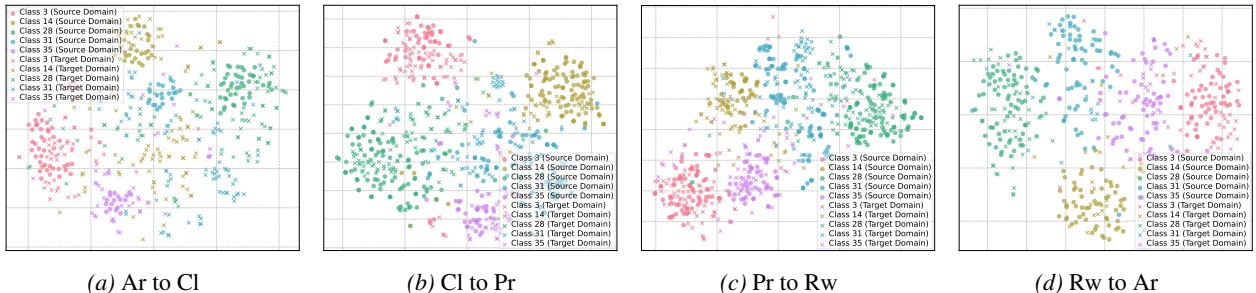

*(a)* Ar to Cl         *(b)* Cl to Pr         *(c)* Pr to Rw         *(d)* Rw to Ar

*Figure E.5. Illustration of cluster feature representations in the SFDA classification task on the Office-Home dataset (Venkateswara et al., 2017). The plots compare source and target feature distributions using t-SNE (Van der Maaten & Hinton, 2008) on a general object classification task. In the figures, points represent source-domain data and crosses represent target-domain data, while different colors indicate different classes. Both source and target inputs are processed through the pre-trained source model, which also serves as the initialization for the target model. Dots represent source data, crosses represent target data, and different colors correspond to five randomly selected classes. In the source domain, clear clusters and well-separated class boundaries are observed. In contrast, target samples show clustering tendencies but are more diversely distributed.*

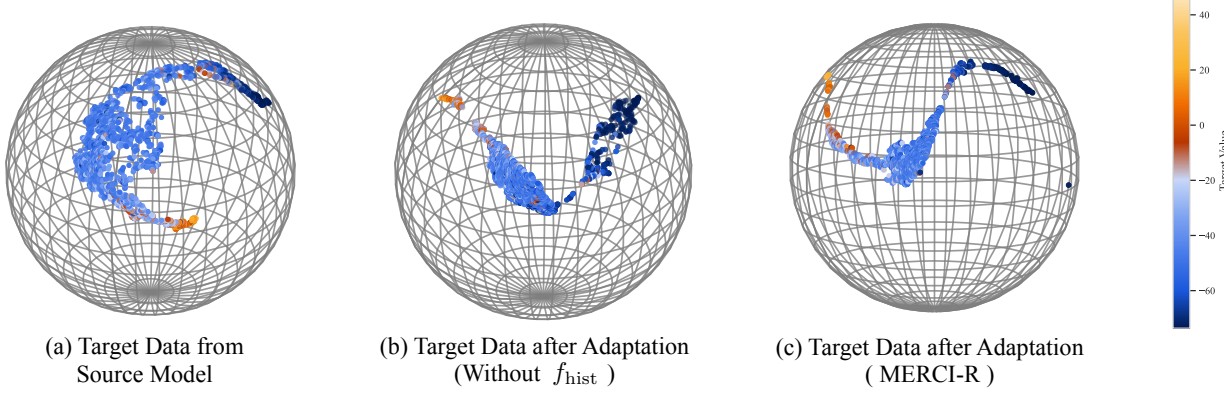

(a) Target Data from
Source Model

(b) Target Data after Adaptation
(Without $f_{\text{hist}}$)

(c) Target Data after Adaptation
( MERCI-R )

*Figure E.6. UMAP visualization of target data feature space on UTKFace dataset*

### E.6. Cluster Property in Feature Space of Classification Problem

In classification, discrete labels naturally guide deep models to learn clustered feature embeddings, a phenomenon aligned with the cluster assumption in classification and semi-supervised learning (Chapelle et al., 2009). In our work, the term "clustered feature space" in the Sec. 1 exactly refers to this phenomenon. More specifically, feature representations learned by deep models in classification tasks are separable by class, i.e., samples from the same class form clusters in embedding space that are separated by low-density boundaries. This phenomenon has been observed across various classification tasks and data modalities, including text topic classification (Lee et al., 2024), speech emotion classification (Kim et al., 2017), and animal image classification (Yang et al., 2022a). It has also been widely observed in SFDA classification benchmarks (Liang et al., 2020; Yang et al., 2021).

In contrast, regression involves continuous targets, encouraging smooth feature-to-output mappings without explicit class separation. As a result, deep regression features typically lie on a continuous manifold and lack the clustering structure seen in classification (Islam et al., 2023).

Our observation of clustered structures in the feature space for classification emerged from empirical experiments (see Figure E.5), and was used to highlight the different feature manifolds observed in regression (see Figures 4 and E.6), which may not form clearly separable clusters and thus introduce unique challenges for pseudo-label refinement.

### E.7. Feature Representation Learning in Deep Regression Models

Due to the page limitation of the main paper, we provide additional observation on feature representations learned by deep regression model in this Appendix.

In Figures 4 and E.6, we visualize three types of target domain feature representations on both UTKFace (Male → Female) and Biwi-Kinect (Female → Male, pitch angle prediction) datasets : **(a)** direct feature representations of target data passed through the pre-trained source model; **(b)** adapted target feature obtained by training with only the conventional regression loss ($\mathcal{L}_{\mathrm{RMSE}}$), using initially collected histogram information as supervision signals; **(c)** adapted target feature under our complete MERCI-R framework.

Comparing (a) with (b) and (c), the adapted features exhibit more locally compact structures, reflecting the controlled conditional entropy $\mathcal{H}(Z|Y)$. However, a closer comparison between (b) and (c) shows that the complete MERCI-R framework, equipped with the histogram head and classification loss, achieves both local compactness and global diversity. By contrast, relying solely on MSE or RMSE loss tend to reduce feature entropy $\mathcal{H}(Z)$, which limits generalizability and degrades performance. When the target values are noisy, as in SFDAR settings, the performance of MSE loss becomes unstable, hindering smooth representation learning (Figure 4(b)) and showing weakness in handling extreme values (Figure E.6(b)).

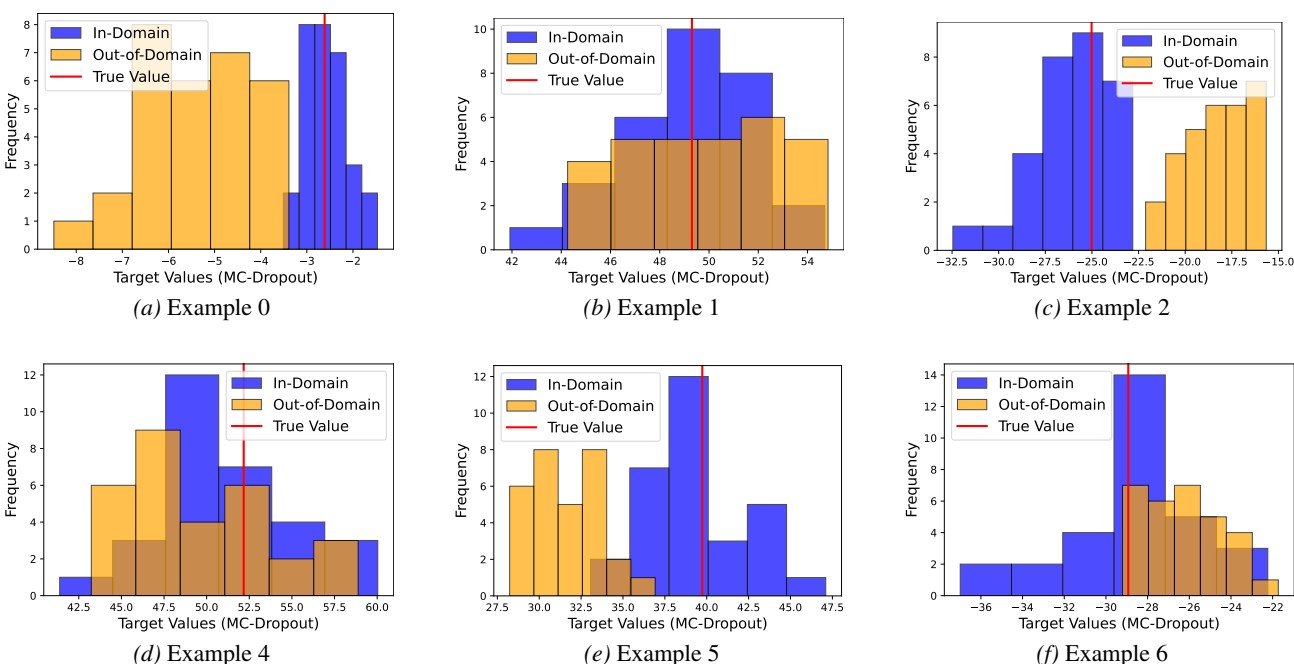

*Figure E.7. MC-Dropout predictive distributions for samples from the female domain on Biwi-Kinect dataset. In-domain predictions (blue) exhibit clear unimodality, while out-of-domain predictions (yellow) deviate from the mean and lack unimodality. These results empirically motivate the use of a Gaussian prior for the histogram head.*

### E.8. Unimodal Conditional Distribution Validation

In this subsection, we explain the unimodality assumption used in MERCI, supported by both theoretical insights and experimental observations.

**Unimodal vs. Multimodal.** When the input $X$ contains sufficient information to predict $Y$, the uncertainty around the prediction is commonly assumed to be dominated by measurement or annotation noise, such as Gaussian or sub-Gaussian behavior. In this setting, the conditional predictive distribution tends to concentrate within a single continuous interval, rather than exhibiting genuinely multimodal structure. Multimodality typically appears when essential information is missing. For example, predicting age from height alone without sex information may lead to two separated plausible ranges. In contrast, given a clear frontal facial image, we intuitively expect age to fall within a contiguous region with high probability instead of forming two distinct clusters, such as around 25 or 45. The target tasks in our experiments, including age estimation on UTKFace and head pose estimation, provide rich visual cues, so the unimodal assumption is a reasonable modeling choice. We also note that our framework can accommodate multimodal scenarios. Specifically, we can (1) model each peak with a separate unimodal component, or (2) incorporate richer prior information describing the multimodal structure and apply it on top of the partial-label set.

**Gaussian prior.** In our experiments, we chose a Gaussian prior for two practical reasons. First, the Gaussian distribution is a standard choice that allows closed-form computation of the likelihood and KL term. Second, when only partial knowledge of a distribution is available (in SFDA setup), selecting a Gaussian prior follows the maximum-entropy principle, meaning that the distribution that best reflects the current state of knowledge is the one with the largest entropy. In particular, the Gaussian maximizes differential entropy under a fixed variance constraint, so it introduces no additional structural assumptions beyond the observed scale of label variability (Rioul, 2021). In our method, dropout sampling together with histogram estimation provides information on the mean and approximate dispersion of the label distribution, making the Gaussian a suitable default in the absence of further empirical evidence. More complex priors are only warranted when such empirical evidence is present. Specifically, if additional structural information about the label distribution becomes available, other priors can be naturally incorporated into our framework. For instance, a Laplace prior may be appropriate when residuals are known to exhibit sharp, heavy-tailed behavior, whereas Gaussian mixtures or non-parametric priors are suitable when multimodality is empirically observed.

**Empirical Evidence.** To examine the unimodality assumption of the conditional distribution $P_{y|x}$, we randomly select several samples from the female domain of the Biwi-Kinect dataset and evaluate them using two models: (1) an in-domain model trained on female-domain data, and (2) an out-of-domain model trained on the male domain. We estimate and visualize the MC-Dropout distributions for each sample to enable a direct comparison and more detailed analysis. As shown in Figure E.7, the MC-Dropout outputs of the well-trained in-domain model exhibit a clear unimodal pattern. In contrast, the out-of-domain model often produces distributions that deviate substantially from the mean, lack unimodality, and display pronounced skewness. These observations empirically motivate the adoption of a Gaussian prior when training the histogram head.

### E.9. Analysis of Mutual Enhancement Between Histogram and Regression Heads

To examine the directional contributions within MERCI, such a mutual-enhancement framework, we conducted an additional set of experiments on the UTKFace dataset. In particular, we selectively disabled two forms of feedback: (1) the feedback from the histogram head to the regression head after the warm-up stage, and (2) the feedback from the regression head to the histogram head after initialization. The resulting MAE values are presented in Table E.15.

As shown in the table, removing the histogram-to-regression feedback leads to a noticeably larger performance degradation, suggesting that this direction plays a more critical role in improving prediction quality. In contrast, removing the regression-to-histogram feedback results in a smaller degradation; however, the initialization of the histogram head still fundamentally relies on the regressor. Therefore, this pathway as well as the overall design of the framework remains unified and mutually consistent.

We further visualize the pseudo-labels quality from the histogram head and the predictions from the regressor at each epoch on Biwi-Kinect (female→male, pitch), shown in Figure E.8. Both outputs show clear descending trends with visible oscillations, demonstrating that the fixed partial-label set and moving average strategy effectively stabilize the noisy pseudo-labels.

*Table E.15. MAE evaluation for directional ablations within the mutual-enhancement framework.*

| Model | Female → Male | Male → Female |
|---|---|---|
| **Full Model** | **5.82** | **7.0802** |
| regression $\not\to$ histogram | 6.0492 | 7.1080 |
| histogram $\not\to$ regression | 6.1304 | 7.1704 |

### E.10. Efficiency and Scalability Analysis

In this section, we provide a detailed efficiency and scalability analysis of our proposed method, MERCI, including (i) initialization time, (ii) per-epoch training time, (iii) peak GPU memory usage, and (iv) parameter count on Biwi-Kinect dataset. All experiments were conducted on a single NVIDIA GPU under the same settings as the main paper for a fair comparison.

**Initialization Time.** MERCI includes a one-time initialization step that involves stochastic forward passes and a density estimation. As shown in Table E.16, the initialization takes about 305 seconds, of which fewer than 10 seconds correspond to KDE estimation. The total initialization cost is comparable to that of TASFAR (around 430 seconds).

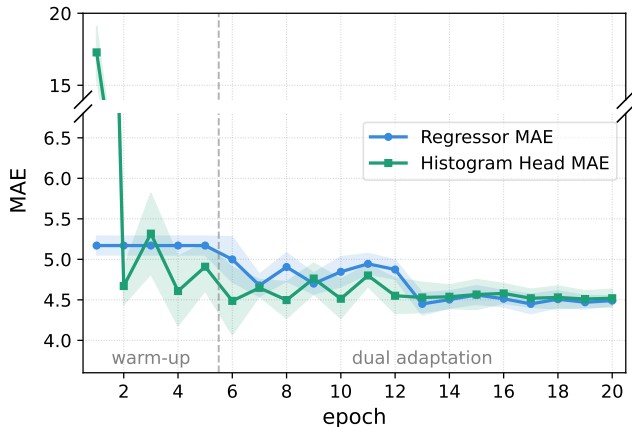

*Figure E.8. Pseudo-label MAE from the histogram head and the predictions from the regressor at each epoch on Biwi-Kinect (female→male, pitch).*

*Table E.16. Efficiency comparison of MERCI and baseline methods.*

| Method | Init Time (s) | Train Time / Epoch (s) | Peak GPU Memory |
|---|---|---|---|
| BN-adapt (Benz et al., 2021) | 0.00 | 170.91 | 0.73 GB |
| RSD (Chen et al., 2021) | 0.00 | 202.96 | 10.66 GB |
| DANN (Ganin et al., 2016) | 0.00 | 194.63 | 10.76 GB |
| DARE-GRAM (Nejjar et al., 2023) | 0.00 | 308.46 | 10.97 GB |
| VM | 0.00 | 240.49 | 21.04 GB |
| SSA (Adachi et al., 2025) | 86.97 | 190.31 | 5.33 GB |
| CRAFT-U (Biswas et al., 2025) | 176.30 | 178.07 | 5.34 GB |
| BRR (Zhan et al., 2025) | 181.28 | 180.38 | 6.29 GB |
| TASFAR (He et al., 2024) | 435.71 | 180.28 | 5.51 GB |
| MERCI-R | 305.32 | 232.94 | 6.41 GB |

**Per-Epoch Training Time.** MERCI's per-epoch training time (about 230s) is on the same order as other source-free and unsupervised domain adaptation baselines, such as SSA (190.31 s), RSD (202.96 s), DANN (194.63 s), DARE-GRAM (308.46 s), VM (240.49 s), and TASFAR (180.28 s). Despite including an initialization stage, the iterative training cost of MERCI remains competitive.

**Peak GPU Memory Usage** We also report peak GPU memory usage in Table E.16. MERCI requires only 6.41 GB, which is substantially lower than traditional unsupervised domain adaptive regression methods such as RSD, DANN, DARE-GRAM, and SFDAR method VM. This demonstrates that MERCI is computationally manageable and memory-efficient, enabling scalability to common backbone architectures.

**Parameter Count.** Finally, Table E.17 summarizes the parameter count. The backbone plus regressor contains about 120.30M parameters, while the proposed histogram head adds only 0.53M (<0.5% overhead), indicating that MERCI introduces minimal and controllable architectural complexity.

### E.11. Calibration Analysis of Source Model on Target Domain

To evaluate how well the uncertainty from the source model transfers to the target domain, we conduct a calibration analysis using two stochastic prediction mechanisms: MC-Dropout (MC-D) and Augmentation Ensemble (Aug-Ens). For each target

*Table E.17. Parameter count of MERCI components.*

| Component | Parameters |
|---|---|
| Backbone + Regressor | 120.30M |
| Histogram Head | 0.53M |

*Table E.18.* Calibration comparison on Biwi-Kinect (female $\rightarrow$ male). MC-D: MC-Dropout; Aug-Ens: Augmentation Ensemble (both from source model on target data, before adaptation); Hist: adapted histogram head output. Target $P$ denotes the nominal CI level; PICP is the empirical coverage; **lower $|\text{PICP}-P|$ indicates better calibration**.

| Variable | Target $P$ | MC-D (before adapt.) | | Aug-Ens (before adapt.) | | Hist (after adapt.) | |
|---|---|---|---|---|---|---|---|
| | | PICP | $|\text{PICP}-P|$ | PICP | $|\text{PICP}-P|$ | PICP | $|\text{PICP}-P|$ |
| pitch | 50% | 0.3006 | 0.1994 | 0.2546 | 0.2454 | 0.4473 | **0.0527** |
| pitch | 60% | 0.3662 | 0.2338 | 0.3147 | 0.2853 | 0.5526 | **0.0474** |
| pitch | 70% | 0.4390 | 0.2610 | 0.3822 | 0.3178 | 0.6461 | **0.0539** |
| pitch | 80% | 0.5211 | 0.2789 | 0.4693 | 0.3307 | 0.7325 | **0.0675** |
| pitch | 90% | 0.6405 | 0.2595 | 0.5793 | 0.3207 | 0.8381 | **0.0619** |
| roll | 50% | 0.1938 | 0.3062 | 0.1643 | 0.3357 | 0.3680 | **0.1320** |
| roll | 60% | 0.2461 | 0.3539 | 0.2070 | 0.3930 | 0.4438 | **0.1562** |
| roll | 70% | 0.2978 | 0.4022 | 0.2622 | 0.4378 | 0.5249 | **0.1751** |
| roll | 80% | 0.3657 | 0.4343 | 0.3231 | 0.4769 | 0.6270 | **0.1730** |
| roll | 90% | 0.4721 | 0.4279 | 0.4273 | 0.4727 | 0.7574 | **0.1426** |
| yaw | 50% | 0.1790 | 0.3210 | 0.1382 | 0.3618 | 0.6004 | **0.1004** |
| yaw | 60% | 0.2296 | 0.3704 | 0.1711 | 0.4289 | 0.6937 | **0.0937** |
| yaw | 70% | 0.2880 | 0.4120 | 0.2078 | 0.4922 | 0.7842 | **0.0842** |
| yaw | 80% | 0.3572 | 0.4428 | 0.2545 | 0.5455 | 0.8797 | **0.0797** |
| yaw | 90% | 0.4562 | 0.4438 | 0.3188 | 0.5812 | 0.9682 | **0.0682** |

sample, multiple stochastic predictions are generated. Under a Gaussian approximation of these sampled predictions, we estimate a predictive mean $\mu_i$ and standard deviation $\sigma_i$, and construct a predictive interval $\hat{I}_i$ at confidence level $P$ as:

$$\hat{I}_i = \left[\, \mu_i - Z_P \sigma_i, \ \mu_i + Z_P \sigma_i \,\right],$$

where $Z_P$ is the standard normal quantile corresponding to confidence level $P$. This Gaussian-based interval approximates the range that the model expects to contain the true value with probability $P$.

To quantify empirical calibration, we use the Prediction Interval Coverage Probability (PICP), defined as the fraction of target samples whose ground-truth values fall within the predictive interval $\hat{I}_i$:

$$\text{PICP} = \frac{1}{N} \sum_{i=1}^{N} \mathbf{1}[\, y_i \in \hat{I}_i].$$

A well-calibrated estimator should satisfy $\text{PICP} \approx P$.

The results in Table E.18 show that MC-D is consistently under-calibrated on the target domain, and Aug-Ens exhibits even stronger under-coverage across all confidence levels and variables. This indicates that the miscalibration arises not from a specific estimator but from the inherent distribution shift between source and target domains. These observations also clarify why uncertainty-dependent SFDA methods such as TASFAR perform poorly, and they motivate MERCI's use of partial label sets and a histogram head to mitigate miscalibrated uncertainty and better capture the target-domain density structure. Empirically, the adapted histogram head achieves substantially better calibration across all three head pose variables, confirming that adaptation improves density reliability beyond merely reducing point estimation error.

