# OpenReview forum: "Discretized Density-Guided Source-Free Domain Adaptation for Regression"
_ICML.cc/2026/Conference — ICML 2026 spotlight_

### Official Review · Reviewer_Netp · 2026-03-09

**Soundness:** 3
**Presentation:** 4
**Significance:** 3
**Originality:** 3
**Overall Recommendation:** 4
**Confidence:** 3

**Summary:**

This paper addresses source-free domain adaptation for regression tasks. Noticing that regression outputs are continuous and deterministic, which limits uncertainty estimation and makes pseudo-label refinement difficult, the paper proposes MERCI, a framework that integrates discretized-density estimation. The approach performs bi-directional knowledge transfer between the regression and histogram head. Experiments on several regression benchmarks show improved performance.

**Compliance With Llm Reviewing Policy:**

Affirmed.

**Final Justification:**

The rebuttal addresses my concerns, therefore maintaining my positive score.

**Key Questions For Authors:**

1. The method assumes the conditional label distribution to be unimodal and models it with a Gaussian prior. Could the authors discuss how the method behaves when the assumption does not hold?
2. To enhance the dropout-based stochastic predictions, is it helpful to apply some diversity-regularization on the network parameters during training?
3. The method selects $K$ bins automatically by the algorithm, however, the label space is still uniformly paritioned. Could the authors discuss what if the label space is imbalanced and skewed? Will a uniform parition be insuffient?

**Limitations:**

Yes

**Strengths And Weaknesses:**

Strengths
- Different from works on commonly studied classification tasks, this paper focuses on the less-studied regression tasks, aiming to address the unique challenges from continous targets, thus is highly original and relevant in domain adaption area.
- The proposed dual-head architecture is well motivated to introduce uncertainty estimation into deterministic regression models. The method is supported by theoretical analysis that discusses the histogram-based density approximation.
- Extensive experiments across multiple regression tasks validate its effectiveness.
- The paper is well structured, with clear formulation, method description, figure illustration, theoretical analysis and experimental results.
- The paper tackles a practical regression problem. The proposed histogram-based uncertainty estimation can be extend to more general regression tasks.

Weaknesses
- The pipeline is relatively complex. The final MERCI loss consists of four terms, making it difficult to select most critical components on new data.
- The unimodal conditional label distributions assumption may not always hold in real-world regression tasks.
- The method uses dropout-based stochastic predictoins. However, for regression tasks, the results are more deterministic than classification tasks. It is helpful to discuss how well the outputs with small variances cover $K$ predefined bins. Would it be possible most of the bins have zero probability mass?
- (minor) some loss definitions are included in appendix, making it unclear when reading the main texts.

---

> ### Author Rebuttal · Authors · 2026-03-31
>
> We thank Reviewer Netp for the valuable feedback. Below, we address each concern in turn.
>
> ---
>
> **Response to W1** (Complex pipeline and loss terms): We would like to clarify that the loss components in MERCI are modular with clearly separated roles.
> - The **KL loss and Partial Label loss** are associated with the histogram head and are always used to learn the discretized density.
> - For the regression and representation learning components:
>     - The **regression loss** can be either MSE or RMSE. In particular, **RMSE is derived from the DRO formulation**, which explicitly accounts for pseudo-label uncertainty and improves robustness.
>     - The **feature norm regularization** is imposed on the feature space to encourage more compact and transferable representations.
> - In practice, we find that RMSE is more beneficial under large target distribution shifts, while feature norm is helpful when the domain gap is significant. As a result, we adopt **RMSE + feature norm** as the default setting, which performs consistently well across datasets.
> Overall, a fixed default configuration (RMSE + feature norm) works well in practice, with learnable weights eliminating manual tuning.
>
> ---
>
> **Response to W2&Q1** (unimodal conditional label distributions assumption):
> - **Justification**. We adopt the unimodal conditional label distribution assumption as it holds in many real-world regression tasks, where each input typically corresponds to a single continuous outcome (e.g., head pose, age). We further provide empirical validation through a case study in Appendix E.7. Importantly, **the unimodal prior is a soft regularizer rather than a hard constraint.** It is incorporated via $L_\text{KL}$ with a learnable weight, automatically balanced with the partial label loss. In the extreme case where  $\lambda_\text{prior} \to 0$, the method becomes fully data-driven and imposes no unimodal assumption.
> - **Failure scenario.** When the assumption is mildly violated, the method remains functional with bounded error. For example, under a weakly bimodal conditional distribution, if the partial label set covers both modes, the learned histogram interpolates between them, and the pseudo-label error is bounded by half the inter-mode distance.
> We thank the reviewer for this insightful comment and acknowledge that strongly multimodal settings represent a genuine limitation and an interesting direction for future work.
>
> ---
>
> **Response to W3**: Due to the relatively small variance of dropout-based predictions in regression, the resulting partial label set typically covers only a local subset of bins, rather than the full discretized space. As shown in [$\color{red}\textbf{Figure}$](https://anonymous.4open.science/r/2026_merci-3494/samples_histogram_pitch.pdf), the initial partial label set is concentrated locally.
> - Although the initial partial label set covers only a limited subset of bins, the learning process of the histogram head, guided by the partial label loss and a Gaussian prior, expands the supported bin range, producing a smoother and more distributed density beyond the initial support.
> - While many bins may have near-zero probability mass, this does not affect performance. What matters is that the **high-probability region, shaped by the partial label set, sufficiently covers the ground truth**, which enables reliable pseudo-label estimation and correction.
>
> ---
>
> **Response to Q2**:
> - We would like to clarify that dropout is used only for **stochastic sampling**, i.e., constructing the partial label set and the histogram, and is **not used during final prediction**, where the regressor head remains deterministic.
> - Applying diversity regularization during adaptation may introduce side effects, as it could interfere with the stability of pseudo-label refinement. However, we find this suggestion promising for encouraging diversity at the **sampling stage**, which could potentially improve the coverage of the partial label set. We will consider this direction in future work.
>
> ---
>
> **Response to Q3**: We thank the reviewer for this insightful suggestion.
> - We have explored an alternative binning strategy based on **quantiles** initially, which results in bins with uniform probability mass but varying widths. However, it introduces instability and limited gains.
> - Moreover, our design choice of **uniform partitioning** is motivated by interpretability: it allows the histogram head to approximate a density with meaningful and consistent support over the label space, rather than adapting bin widths to the empirical distribution.
> - We agree that non-uniform binning could be beneficial for highly imbalanced or skewed label distributions, and may also impact feature representation learning. We consider this a promising direction for future work.
>
> ---
> We thank Reviewer Netp again for the helpful comments and will revise the manuscript accordingly, including reorganizing the loss definitions and incorporating the additional discussion.

---

> > ### Author Rebuttal · Reviewer_Netp · 2026-04-03
> >
> > I appreciate the response. It solves my concerns. I would like to keep the acceptance.

---

> > > ### Author Response · Authors · 2026-04-05
> > >
> > > Dear Reviewer Netp,
> > >
> > > We are very pleased to hear that your concerns have been addressed. Thank you for your thoughtful feedback and for your kind acknowledgment of our response. Your comments have been very helpful in improving the clarity and quality of our work.
> > >
> > > We truly appreciate your support and your decision to maintain your positive rating.
> > >
> > > Sincerely,
> > >
> > > The Authors

---

### Official Review · Reviewer_8Dtj · 2026-03-10

**Soundness:** 2
**Presentation:** 1
**Significance:** 2
**Originality:** 2
**Overall Recommendation:** 4
**Confidence:** 3

**Summary:**

This paper tackles the complex problem of Source-Free Domain Adaptation (SFDA) with a specific focus on continuous regression tasks, differentiating it from the more heavily researched classification domain. The proposed methodology introduces a histogram-based discretization approach to manage the regression space, paired with pseudolabeling and distributionally robustness. The authors support their framework with detailed derivations and comprehensive empirical evaluations across four distinct datasets.

**Compliance With Llm Reviewing Policy:**

Affirmed.

**Final Justification:**

The authors have addressed my concerns in the rebuttal response, and I have raised my score from 3 to 4 accordingly.

**Key Questions For Authors:**

- Could you elaborate on the specific algorithmic and theoretical challenges of applying SFDA to regression as opposed to classification, and how your method explicitly addresses them?
 - What are the primary differences between your histogram-based discretization methodology and existing discretization methods in the literature, such as BRR (Zhan et al.)?
 - How does the framework handle non-overlapping source and target label spaces? Does the histogram-based approach inherently limit applicability in these scenarios?
 - The loss term weightings (λ) are learnable. What specific mechanisms prevent these values from decaying towards zero during training and trivially minimizing the loss?
 - Can this methodology be seamlessly extended to a setting with multiple target domains, or is it strictly constrained to single-target adaptation?

**Limitations:**

yes

**Strengths And Weaknesses:**

### Strengths:

 - Thorough Analytical Approach: The paper provides a highly detailed and rigorous theoretical foundation for its proposed methodology.

 - Comprehensive Empirical Evaluation: The method is evaluated across four separate datasets, demonstrating a solid effort to validate the framework's performance across diverse regression scenarios.

### Weaknesses:

 - Bloated and Dense Presentation: Section 4 suffers from an overabundance of notation. This makes the core methodological contributions difficult to digest and negatively impacts the paper's overall readability.

 - Training on pseudolabels in a source-free regression context is highly susceptible to noise accumulation and error propagation. How does the proposed method address this issue?

 - The authors mention uncertain-awareness as an advantage, but there is no evaluation regarding this uncertainty estimation.

 - Insufficient Benchmarking: The evaluation is missing tests on large-scale benchmarks, which questions the method's scalability and real-world robustness.

 - Incomplete Efficiency Metrics: The paper lacks a comparison of parameter counts against relevant baselines, making it difficult to judge the trade-off between regression performance and model efficiency.

---

> ### Author Rebuttal · Authors · 2026-03-31
>
> We thank Reviewer 8Dtj for the valuable feedback. Below, we address each concern in turn.
>
> ---
> Response to W1: We clarify that the notation in Section 4 is inherently dense, as our framework jointly models continuous targets and their discretized histogram representations. To improve readability, we will:
> 1. move the notation table (Table B.1) from the Appendix to the main paper for quick reference;
> 2. summarize the notation patterns at the beginning of Section 4. For example, the superscript b (e.g., $\hat{y}^{\text{b}}_i$) denotes discretized labels, while variables without superscripts represent continuous values.
>
> ---
>
> Response to W2: Our method mitigates error propagation from two aspects: (1) a **partial-label strategy**, which provides broader coverage of true target values and reduces the impact of incorrect pseudo-label assignments; and (2) a **moving-average update**, which stabilizes training and suppresses noise accumulation. The effectiveness of these designs is validated in Figure 3, Figure E.3, and Table E.5. Please refer to our response to Q1 (Reviewer nGm7) for more details.
>
> ---
>
> Response to W3:
> - In our framework, uncertainty is captured by two components: (1) the stochastic forward process (via MC-Dropout), which models predictive variability, and (2) the histogram head, which provides an explicit probabilistic representation of the target density. The calibration evaluation of the first component is provided in Appendix E.10.
> - Additonal calibration analysis on the histogram head output ([$\color{red}\textbf{Table}$](https://anonymous.4open.science/r/2026_merci-3494/cali.pdf)) further confirms improved density reliability.
>
> ---
>
> Response to W4: We have added experiments on MPI3D; please see our response to Reviewer 3HBK (Limitations) for details.
>
> ---
>
> Response to W5: The efficiency and scalability analysis **are provided in main text (Lines 415–417) and Appendix E.9 (Lines 1551–1594)**. As shown in Table E.13, the additional parameters introduced by the histogram head are negligible compared to the baselines (backbone and regressor).
>
> ---
>
> Response to Q1:
> The key challenges of SFDA for regression are twofold:
> - **No discrete label structure.** Classification naturally supports pseudo-label thresholding, prototype alignment and entropy minimization over a finite label space. In regression, the continuous label space makes reliable supervision construction non-trivial. MERCI uses a **learned** histogram density to enable pseudo-label construction under KL regularization.
> - **Miscalibrated uncertainty.** Softmax scores provide natural confidence proxies in classification, while regression point estimates do not. MERCI combines MC-Dropout with the learned histogram density to produce soft, uncertainty-aware pseudo-labels.
>
> ---
>
> Response to Q2: Here are two key differences from BRR:
> - **Objective.** BRR applies discretization implicitly as a heuristic for bias reduction, without modeling the label distribution. MERCI instead explicitly models the **conditional label distribution**, enabling uncertainty-aware supervision via the histogram head.
> - **Mechanism.** (1) BRR uses a **fixed** discretization and computes pseudo-labels via a heuristic feature-similarity algorithm. (2) MERCI instead **self-adaptively** initializes the histogram structure from target data; during adaptation, the **density** (probability mass over bins) is learned through partial label supervision and KL regularization, replacing hand-crafted heuristics with a principled learning objective.
>
> ---
>
> Response to Q3: We thank the reviewer for this insightful question.
> - To address this concern, we evaluate on UTKFace with **disjoint label spaces** (Female: y < 35, Male: y > 35).
> - As shown in [$\color{red}\textbf{Table}$](https://anonymous.4open.science/r/2026_merci-3494/8Dtj_Q3.pdf), MERCI-R consistently outperforms SFDA baselines (BN-Adapt and BRR) in both directions, as the histogram is built from **target-domain stochastic predictions** that enable uncertainty-aware extrapolation beyond the source label range or fixed discretization.
> - These results confirm that the histogram-based design is (even more) effective in non-overlapping settings.
>
> ---
>
> Response to Q4: To avoid the trivial solution such that λ $\to$ 0, we follow a well-established uncertainty-based multi-task weighting formulation. Please refer to Appendix E.3 (Lines 1274-1306) for more details.
>
> ---
>
> Response to Q5: Our current formulation follows the standard single-target SFDAR setting. Extending to multiple targets is feasible in principle but requires additional cross-target modeling mechanisms, which is largely orthogonal to our contribution. We believe that MERCI's uncertainty-aware, histogram-based supervision can be naturally integrated into future multi-target adaptation frameworks.
>
> ---
> We thank Reviewer 8Dtj again for the constructive comments and hope the above clarifications address the concerns. We look forward to further discussion.

---

> > ### Author Rebuttal · Reviewer_8Dtj · 2026-04-01
> >
> > Most concerns are resolved, and I have raised by score accordingly.

---

> > > ### Author Response · Authors · 2026-04-05
> > >
> > > Dear Reviewer 8Dtj,
> > >
> > > We are very glad to hear that most of your concerns have been addressed and sincerely appreciate the updated score. Thank you for your thoughtful feedback and for engaging with our responses so carefully. Your comments have been very helpful in improving the clarity and quality of our work.
> > >
> > > We would be happy to continue addressing any further questions or suggestions that may arise.
> > >
> > > Sincerely,
> > >
> > > The Authors

---

### Official Review · Reviewer_nGm7 · 2026-03-12

**Soundness:** 4
**Presentation:** 4
**Significance:** 4
**Originality:** 4
**Overall Recommendation:** 5
**Confidence:** 5

**Summary:**

In this paper, the authors introduces a histogram head that reformulates the continuous regression into a discretized density (histogram) estimation problem. By mapping continuous ground-truth values to the discretized bins, the proposed method offers a structured signal that can refine the pseudo-labels through truncated expectations. The proposed work effectively allows the regression models to benefit from the clustering-based feature learning methods typically used in the classification-based DA methods. Moreover, this classification-style head allows the use of entropy minimization and KL-divergence to capture and regularize prediction uncertainty in the target domain.

**Compliance With Llm Reviewing Policy:**

Affirmed.

**Final Justification:**

The proposed work is well-motivated and both theoretically and empirically supported. It makes it possible to apply the classification-based DA methods directly to the regression-based DA problems via a self-adaptive histogram discretization. I find this paper to be very interesting and promising.

Meanwhile, all my concerns are properly addressed during the rebuttal. The authors provide results for all suggested ablation experiments. Thus, I am very happy to maintain my positive rating, i.e., Accept.

**Key Questions For Authors:**

1. The calibration analysis demonstrates that the MC-Dropout is a bit under-calibrated on the target domain due to the domain shifts. This could imply that the histogram head's refinement might lead to the noisy pseudo-labels that could negatively affect the adaptation. I would suggest the authors to conduct more ablation studies to address such concerns to make the proposed work more impactful.

2. While the authors demonstrate the stability across a wide range of experiments, the model's performance might degrade if the bin count is extremely small. Could the authors conduct an ablation experiment to alleviate such concerns?

**Limitations:**

yes

**Strengths And Weaknesses:**

**Strengths:**

1. The proposed work bridges the gap between classification and regression DA by using a dual-head architecture that allows regression tasks to leverage the uncertainty-aware signals typically proposed in the classification-based DA methods.

2. By introducing a dropout sampling, the proposed work addresses the limitations of the regression-based DA methods by offering a quantitative measure of the prediction confidence using entropy.

3. The authors introduce a theoretically-grounded and empirically-supported method to determine the number and the size of the discretized bins based on the local density of the target distribution.

4. The UMAP visualizations demonstrate that the proposed losses can also improve the feature extraction by making the latent features more compact and structural.

**Weaknesses:**

1. The calibration analysis demonstrates that the MC-Dropout is a bit under-calibrated on the target domain due to the domain shifts. This could imply that the histogram head's refinement might lead to the noisy pseudo-labels that could negatively affect the adaptation. I would suggest the authors to conduct more ablation studies to address such concerns to make the proposed work more impactful.

2. While the authors demonstrate the stability across a wide range of experiments, the model's performance might degrade if the bin count is extremely small. Could the authors conduct an ablation experiment to alleviate such concerns?

Overall, the proposed work is well-motivated and both theoretically and empirically supported. It makes it possible to apply the classification-based DA methods directly the regression-based DA problems via a self-adaptive histogram discretization. Meanwhile, the authors provide solid solutions for pseudo-label refinement and representation learning that outperforms other baselines across diverse benchmarks. I find this paper to be very interesting and promising.

---

> ### Author Rebuttal · Authors · 2026-03-31
>
> We thank Reviewer nGm7 for the valuable feedback. Below, we address each question in turn.
>
> ---
>
> **Response to W1 & Q1:** We thank the reviewer for raising this important point. We address this concern as follows:
> 1. From a methodology perspective, MERCI is designed to reduce this risk in two ways.
>     - First, we do not use MC-Dropout outputs (e.g., mean predictions) as hard labels. Instead, we construct a partial label set, which provides broader supervision coverage (as shown in Table 3 in the manuscript), so that errors in pseudo-labels do not immediately lead to incorrect hard assignments.
>     - Second, we use a moving-average update to smooth the adaptation process, which helps reduce unstable pseudo-label updates and mitigates error propagation.
> 2. From the experimental perspective, the effects of the **partial-label set** and **moving-average strategy** are analyzed in **Figure 3**, **Figure E.3**, and **Table E.5**, with pseudo-label quality further summarized in **Table 3** (Partial Label Set Quality and Continuous Pseudo-Label Quality columns). To provide a more intuitive understanding of pseudo-label noise, we offer two additional analyses:
>     - **Point estimation accuracy.** We evaluate the pseudo-labels from the histogram head and the predictions from the regressor at each epoch on Biwi-Kinect (female→male, pitch), shown in [$\color{red}\textbf{Figure}$](https://anonymous.4open.science/r/2026_merci-3494/dual_adaptation_curve.pdf). Both outputs show clear descending trends with visible oscillations, demonstrating that the fixed partial-label set and moving average strategy effectively stabilize the noisy pseudo-labels.
>     - **Calibration.** We evaluate the **calibration of the histogram head** using the PICP metric (introduced in Appendix E.10) across nominal confidence levels (50%–90%). As shown in the [$\color{red}\textbf{Table}$](https://anonymous.4open.science/r/2026_merci-3494/cali.pdf), the source model's MC-Dropout and augmentation ensemble estimates are severely miscalibrated on the target domain. In contrast, the adapted histogram head achieves substantially better calibration across all three head pose variables, confirming that adaptation improves density reliability **beyond merely reducing point estimation error**.
> 3. We acknowledge that noisy pseudo-labels remain a limiting factor in SFDAR problem. Building on the uncertainty-aware outputs of the histogram head in MERCI, we believe that incorporating uncertainty-based sample selection is a promising direction for future work.
>
> ---
>
> **Response to W2 & Q2 (extremely small bin number):**
>
> **Possible failure discussion.** We thank the reviewer for this comment. When the bin count is extremely small, the histogram becomes too coarse to represent the label distribution, leading to large quantization error. This degrades the quality of the partial-label supervision and KL regularization, and may propagate inaccurate pseudo-labels to the regressor, resulting in performance collapse.
>
> **Experimental validation.**
> To verify this, **we extend the ablation in Table E.8 to a wider range of K values** (note: the leftmost point in Table E.8 should be K=15, not K=5; we will correct this typo in the revised manuscript). We report both the final MAE and the **Histogram Head GT MAE** (oracle upper bound) in Table 1. The results show that when K<10, large discretization error lead to performance collapse. As K increases beyond 15, both metrics stabilize, confirming that the model is robust over a wide range of bin counts. Besides, our self-adaptive K effectively avoids the low-K failure regime by selecting a data-driven bin count, achieving competitive performance without manual tuning.
>
> We will add a dedicated discussion of this failure mode to the limitation section of the revised manuscript, to provide practical guidance for users of our method.
>
> **Table 1: Ablation study on bin number.**
>
> | Bin Number                   | 3     | 5     | 10   | 15   | 25   | 45   | 65   | 85   | 100  | Self-Adaptive |
> | ---------------------------- | ----- | ----- | ---- | ---- | ---- | ---- | ---- | ---- | ---- | ------------- |
> | UTKFace (female→male)<br>MAE | 23.98 | 14.54 | 7.92 | 6.55 | 5.97 | 5.82 | 5.88 | 5.83 | 5.84 | 5.83          |
> | Histogram Head GT<br>MAE     | 23.65 | 14.69 | 6.51 | 4.74 | 2.74 | 1.73 | 0.91 | 0.70 | 0.70 | 1.84          |
> | UTKFace (male→female)<br>MAE | 17.09 | 12.74 | 8.66 | 7.94 | 7.05 | 7.06 | 7.08 | 7.33 | 7.00 | 7.08          |
> | Histogram Head GT<br>MAE     | 18.09 | 13.16 | 6.82 | 4.37 | 2.78 | 1.10 | 0.87 | 0.78 | 0.69 | 0.87          |
>
> ---
> We thank Reviewer nGm7 again for the insightful questions and constructive comments, and hope the above clarifications address the concerns. We look forward to further discussion.

---

> > ### Author Rebuttal · Reviewer_nGm7 · 2026-04-02
> >
> > Thank the authors for providing the results for the suggested ablation experiments. All my concerns are properly addressed. I am very happy to maintain my positive rating.

---

> > > ### Author Response · Authors · 2026-04-05
> > >
> > > Dear Reviewer nGm7,
> > >
> > > We are very pleased to hear that our rebuttal has adequately addressed your concerns. We sincerely appreciate the time and effort you invested in reviewing our work, as well as your constructive suggestions throughout the process. Your feedback has been invaluable in improving the paper.
> > >
> > > Thank you again for your kind recognition and support.
> > >
> > > Sincerely,
> > >
> > > The Authors

---

### Official Review · Reviewer_3HBK · 2026-03-12

**Soundness:** 3
**Presentation:** 3
**Significance:** 3
**Originality:** 2
**Overall Recommendation:** 4
**Confidence:** 3

**Summary:**

This paper investigates the problem of source-free domain adaptation (SFDA) in regression tasks. A Discrete Density-Guided Source-Free Adaptation (DDG-SF) framework is proposed. Its core idea is to discretize the continuous target distribution into several density-aware bins and utilize this auxiliary density structure to guide pseudo-label refinement. This method combines uncertainty-aware pseudo-label learning with discretized distribution supervision, thereby encouraging the learning of more structured feature representations to improve the model's robustness to distribution shifts.

**Compliance With Llm Reviewing Policy:**

Affirmed.

**Final Justification:**

The author's response has resolved my concerns. Given the paper's innovativeness and theoretical contributions within the SFDAR setting, we consider the revised manuscript to meet the conference requirements.

**Key Questions For Authors:**

(1) How robust is the method with large domain offsets?
(2) What is the computational overhead of the discrete density modeling module?

**Limitations:**

The paper should discuss more explicitly the following limitation: scalability on large-scale datasets.

**Strengths And Weaknesses:**

Strengths ：
（1）Most existing SFDA research focuses on classification tasks, while relatively few studies address regression tasks. Solving the domain adaptation problem for continuous objectives without source data is of great significance and has strong practical application value.

（2）The paper provides a theoretical argument demonstrating that the discretized density structure is robust to disturbances, thus attempting to explain the source of the method's stability.

Weaknesses
（1）Limited in innovation, the main technical components of this method are mostly derived from existing ideas, making it more like a combination of existing technologies than a completely new approach.
（2）The article does not consider the information loss caused by discretization, nor does it provide a theoretical analysis of discretization.
(3) Although the paper provides some theoretical discussion on the robustness of density structures, the overall analysis remains rather superficial.

---

> ### Author Rebuttal · Authors · 2026-03-31
>
> We thank Reviewer 3HBK for the valuable feedback. Below, we address each concern in turn.
>
> ---
>
> **Response to W1:** We clarify our technical novelty as follows:
> - **Framework Novelty.**  SFDAR remains underexplored with limited established solutions, largely due to the challenges of adapting regression models without source data or reliable supervision. To address this, MERCI proposes a novel **dual-head framework** that dynamically couples regression refinement, histogram density learning, and feature representation compression.
> - **Technical Differentiation.**  Unlike prior methods that rely on fixed or heuristic discretization (e.g., BRR), MERCI **adaptively constructs an instance-dependent density from model uncertainty and learns it under Partial Label (PL) and KL losses**, forming a principled distributional supervision. This learned density further **interacts with the regression head through bi-directional refinement**, enabling uncertainty-aware supervision and more reliable adaptation under distribution shifts.
>
> ---
>
> **Response to W2:** We clarify the impact of discretization as follows:
> - **Controlled discretization error.**  Discretization introduces a quantization error $|\hat{y}_m - \hat{y}^b_m| < \frac{\Delta}{2}$, where the bin width $\Delta = \frac{\text{range}(\mathcal{Y})}{K}$ is adaptively controlled (Appendix C.1). In practice, this error is small compared to the **overall adaptation error**. For example, on Biwi-Kinect, the quantization error (1.42) is substantially smaller than the adaptation MAE (4.3), indicating that discretization is not the primary performance bottleneck. The full statistics across all datasets are provided in [$\color{red}\textbf{Table}$](https://anonymous.4open.science/r/2026_merci-3494/3HBK_W2.pdf).
> - **Not the final prediction.**  We clarify that the discretized density is only an intermediate representation, and the final pseudo-label is computed via the truncated weighted expectation (Eq. (4)), mapped back to the continuous space, recovering the lost precision.
>
> ---
>
> **Response to W3:** We clarify that our theoretical contributions are tightly coupled with the design of MERCI and address key aspects of SFDAR, rather than general regression settings, including learning under noisy supervision and distribution shift. Our analysis is developed from two complementary perspectives.
> - First, since MERCI learns a **histogram head** to estimate the conditional label density from imperfect pseudo-labels, Proposition 4.1 (grounded in Legendre–Fenchel duality) provides a robustness guarantee for the learned density structure.
> - Second, Section 4.2 analyzes the transfer robustness of the **regressor**, characterizing how the training objective behaves under domain shift.
> We acknowledge that finer-grained error propagation analysis remains open and will be a promising direction for future work.
>
> ---
>
> **Response to Q1:** The performance of MERCI under varying domain shifts is reported in the **main paper (Lines 375–377) and Appendix E.4**, where we evaluate both a toy dataset and UTKFace with increasing Gaussian noise to simulate different shift severities.
> We additionally include two SFDA baselines (BN-Adapt and BRR) on UTKface robustness ablations. Results ([$\color{red}\textbf{Table}$](https://anonymous.4open.science/r/2026_merci-3494/3HBK_Q1.pdf)) show that (1) MERCI consistently outperforms BRR across different shift levels, demonstrating more stable performance under domain shifts; (2) combining MERCI with BN-Adapt yields further improvements, indicating that our method is complementary to statistics-based feature adaptation and remains robust under larger domain shifts.
>
> ---
>
> **Response to Q2:** The computational overhead is reported in the **main paper (Lines 415–417) and Appendix E.9**, evaluated on the Biwi-Kinect dataset.
> Specifically, the histogram head adds negligible parameters (<0.5% of backbone plus regressor), with training cost and GPU memory comparable to or lower than existing baselines.
>
> ---
>
> **Response to limitations:**
> - **Overall Analysis.** MERCI’s model size and training cost are **comparable to existing baselines** (Sec. 4 and Appendix E.9). The only additional overhead arises from **stochastic forward passes**, which scale linearly with the number of samples and can be efficiently parallelized on modern hardware.
> - **Experimental Validation.** We further evaluate MERCI on the larger-scale MPI3D dataset. The results ([$\color{red}\textbf{Table}$](https://anonymous.4open.science/r/2026_merci-3494/mpi3d.pdf)) show that MERCI generalizes well to larger datasets, with **a moderate and acceptable increase in computational cost**.
> - **Discussion.** We agree that reducing sampling cost is an important direction for future work and will include this discussion in the revision.
>
> ---
>
> We thank Reviewer 3HBK again for the constructive comments and hope the above clarifications address the concerns. We look forward to further discussion.

---

### Decision · Program_Chairs · 2026-04-30

**Decision:**

Accept (spotlight)

**Comment:**

This paper introduces a histogram head that reformulates the continuous regression into a discretized density (histogram) estimation problem. By mapping continuous ground-truth values to the discretized bins, the proposed method offers a structured signal that can refine the pseudo-labels through truncated expectations.

After the discussions, all concerns are clarified by the authors. Thus, I recommend accepting this paper for this round.